# A thermally activated and highly miscible dopant for n-type organic thermoelectrics

Chi-Yuan Yang[1], Yi-Fan Ding[1], Dazhen Huang[2], Jue Wang[1], Ze-Fan Yao [1], Chun-Xi Huang[1], Yang Lu[1], Hio-Ieng Un[1], Fang-Dong Zhuang[1], Jin-Hu Dou [1], Chong-an Di [2], Daoben Zhu[2], Jie-Yu Wang[1], Ting Lei [3] & Jian Pei[1✉]

N-doping plays an irreplaceable role in controlling the electron concentration of organic semiconductors thus to improve performance of organic semiconductor devices. However, compared with many mature p-doping methods, n-doping of organic semiconductor is still of challenges. In particular, dopant stability/processability, counterion-semiconductor immiscibility and doping induced microstructure non-uniformity have restricted the application of n-doping in high-performance devices. Here, we report a computer-assisted screening approach to rationally design of a triaminomethane-type dopant, which exhibit extremely high stability and strong hydride donating property due to its thermally activated doping mechanism. This triaminomethane derivative shows excellent counterion-semiconductor miscibility (counter cations stay with the polymer side chains), high doping efficiency and uniformity. By using triaminomethane, we realize a record n-type conductivity of up to 21 S cm$^{-1}$ and power factors as high as 51 μW m$^{-1}$ K$^{-2}$ even in films with thicknesses over 10 μm, and we demonstrate the first reported all-polymer thermoelectric generator.

[1] Beijing National Laboratory for Molecular Sciences (BNLMS), Key Laboratory of Polymer Chemistry and Physics of Ministry of Education, Center of Soft Matter Science and Engineering, College of Chemistry and Molecular Engineering, Peking University, Beijing 100871, China. [2] Beijing National Laboratory for Molecular Sciences (BNLMS), CAS Key Laboratory of Organic Solids, Institute of Chemistry, Chinese Academy of Sciences, Beijing 100190, China. [3] Department of Materials Science and Engineering, College of Engineering, Peking University, Beijing 100871, China. ✉email: jianpei@pku.edu.cn

Organic semiconductors have attracted broad attention due to their tunable optoelectronic property, high mechanical flexibility, and good solution processability[1]. Chemical doping is one of the most efficient approaches to filling electron traps, controlling charge carrier concentration, and tuning Fermi level position of the organic semiconductors[2]. Doping has been widely used to improve the performance of organic semiconductor devices, including organic field-effect transistors, organic photovoltaic cells, organic light-emitting diodes, and organic thermoelectric generators[3–5]. P-doping with Lewis acid or F4TCNQ enhanced electrical conductivity of poly[2,5-bis(3-alkylthiophen-2-yl)thieno(3,2-b)thiophene] (PBTTT) to $250 \sim 1100\, \mathrm{S\,cm^{-1}}$[6–8]. Doped poly(3,4-ethylenedioxythiophone) (PEDOT) materials exhibited p-type electrical conductivity over $1500\, \mathrm{S\,cm^{-1}}$, with thermoelectric power factors of up to $469\, \mu\mathrm{W\,m^{-1}\,K^{-2}}$, which demonstrate their application potential compared with commercial inorganic counterparts[9–12]. However, compared with many mature p-doping methods, n-doping is still of challenges, particularly in balancing dopant stability, dopant-semiconductor processability/miscibility, and doping efficiency[2,13,14]. For example, strong electron donors such as alkali metals were initially used for n-doping, but the doping state is instable even under noble gas[15]. Inorganic and organic salts are only processed in vacuum to n-dope organic semiconductors[16–19]. Organic dimer dopants and hydride dopants show good solid-state air stability, solution processability, and strong doping ability[20–22]. For instance, 1,3-dimethyl-2-phenylbenzimidazoline (DMBI)-based hydride dopants can even n-dope p-type materials such as 6,13-bis(triisopropylsilylethynyl)pentacene (TIPS-Pentacene)[23]. However, these dopants still lack solution stability and miscibility with organic semiconductors[11,24]. To overcome the immiscibility, modification of semiconductor backbone using polar side chain or twisted conjugated building blocks are developed; however, these strategies usually significantly change the charge transport property of organic semiconductors, e.g., leading to much lower charge carrier mobility[25–28]. Therefore, a simple and efficient approach is desired to solve the stability and miscibility issues.

Herein, with computer-assisted screening, we develop a triaminomethane-based dopant, namely TAM, to solve the dopant stability and dopant-semiconductor miscibility issues. TAM displays an n-doping ability as excellent as the state-of-the-art n-dopant N-DMBI, whereas it shows much better solubility and chemical stability even in protic solvent. With simple alkyl-substituted guanidine structure, TAM exhibits good counterion-polymer side-chain miscibility, uniform-doping microstructures, and negligible impacts on polymer $\pi$–$\pi$ stacking and charge transport properties. As a result, TAM-doped polymer films can be fabricated as thick as $10\, \mu\mathrm{m}$ without significant sacrificing on device performance: the highest conductivity reaches $21\, \mathrm{S\,cm^{-1}}$ and the power factor reaches $51\, \mu\mathrm{W\,m^{-1}\,K^{-2}}$. To the best of our knowledge, these are the highest values among all reported n-type polymers[29,30]. Furthermore, we demonstrate the first all-polymer flexible thermoelectric generator with an efficient power output of 77 nW in the air.

## Results
**Computer-assisted dopant screening.** Unlike electron donors that offering simple electron, hydride dopants offer hydride ($\mathrm{H^-}$) or a couple of hydrogen atom ($\mathrm{H^\bullet}$) and electron ($\mathrm{e^-}$) in n-doping reactions[31]. The difference in doping mechanism indicate that a high-lying highest occupied molecular orbital is not necessary but a nucleophilic hydrogen is essential in hydride dopant. Therefore, good air stability and strong doping ability can be achieved at the same time[2]. To efficiently offering $\mathrm{H^-}$ (or a couple of $\mathrm{H^\bullet}$ and $\mathrm{e^-}$), the cations of reported hydride dopant are both stabilized by large aromatic rings (Supplementary Fig. 1). After n-doping, the strong

$\pi$–$\pi$ interaction between dopant cations and organic semiconductor anions cannot be ignored, which affects organic semiconductor backbone $\pi$–$\pi$ stacking and charge transport[32,33]. Furthermore, the difference in chemical structure and polarity between organic semiconductor and dopant cations usually cause immiscibility and microphase separation[34,35]. To reduce this cation-semiconductor $\pi$–$\pi$ interaction, we design a new triaminomethane derivatives (TAMs). TAMs as hydride dopants show their $\mathrm{H^-}$ donating properties similar with other hydride n-dopants, such as leuco-pyronin B (LPB)[18,36], leuco-crystal violet (LCV)[37], N-DMBI, and DPDPH[38,39] (Supplementary Figs. 1 and 2), with the active C–H bonds for the nucleophilic hydrogens (Fig. 1a). Moreover, the guanidine-type TAMs$^+$ cations show very small molecular volume and are stabilized by only four-atom Y-aromatic system instead of benzene-based large aromatic rings. To enhance structure and polarizability similarity between TAMs$^+$ and alkyl side chain of semiconductors, alkyl substitutions on nitrogen atoms of TAMs are also designed. As the substituted alkyl type (open-chain or ring-type), steric hindrance and ring tension will affect TAMs' hydride activity, we designed various alkyl substitutions for TAMs, including linear/branched alkyls, cyclic alkyls, fused ring alkyls, and some combinations of rings and linear alkyl groups (Fig. 1b, Supplementary Fig. 2, and Supplementary Note 1).

Density functional theory (DFT) calculations were performed to search for the optimal alkyl substitutions and predict the n-doping ability of TAMs. As n-doping mechanism of hydride dopants is complicated, both hydride-transfer doping and hydrogen-atom-transfer doping pathways can have their contributions[31,39]. The C–H bond strength and negative charge property of active hydrogen would also have influence on the hydride offering ability[40–43]. Therefore, we chose following terms to evaluate their n-doping ability: Gibbs free energy change of the hydride-transfer half-reaction ($\Delta G_{\mathrm{H}^-}$) or hydrogen-atom-transfer half-reaction ($\Delta G_{\mathrm{H}^\bullet}$), singly occupied molecular orbital (SOMO) level of the dopant radical, Mulliken charge, or $^1$H-NMR chemical shift of hydrogen, and the C–H bond length ($d_{\mathrm{C–H}}$), shown in Fig. 1c, d and Supplementary Fig. 2. For an efficient hydride dopant, it is expected to have a weak C–H bond with long $d_{\mathrm{C–H}}$, an electron-rich hydrogen with low Mulliken charge or $^1$H-NMR chemical shift, as well as a small $\Delta G_{\mathrm{H}^-}$ or $\Delta G_{\mathrm{H}^\bullet}$ with high-lying SOMO level[43]. The molecular structures of dopants, dopant radicals, and dopant cations are first optimized, and then the five evaluation factors are calculated (Supplementary Tables 1 and 2) and displayed in the radar charts (Fig. 1c and Supplementary Fig. 2). Some reported dopants (LCV, LPB, DPDHP, and N-DMBI) are also calculated for comparison (Fig. 1d). DFT calculation shows that TAMs with 1,5,7-triazabicyclo[4.4.0]decane backbone (TAM, TAM$_{566}$, TAM$_{667}$, and TAM$_{Me66}$) could be the best n-dopants among all TAMs, and their predicted n-doping abilities are comparable to N-DMBI. This could be explained by the stereoelectronic effect: the fused six-membered rings can form chair conformations in these TAMs and assist the secondary orbital interactions between the C–H anti-bonding orbital and the nitrogen lone pair orbitals (Fig. 1e, f and Supplementary Fig. 3). The secondary orbital interactions weaken the C–H bond, elongate its bond length, enrich electron density of the hydrogen atom, and even make negative electrostatic potential on the hydrogen atom. Moreover, the fused six-membered rings form the planar conformations in these TAM cations, which further stabilize guanidine cations. On the contrary, other TAMs are hard to have the above secondary orbital interactions due to their molecular conformations. Therefore, TAM and other TAMs with 1,5,7-triazabicyclo[4.4.0] decane backbone can balance all the evaluation factors and they are expected to be strong hydride n-dopants.

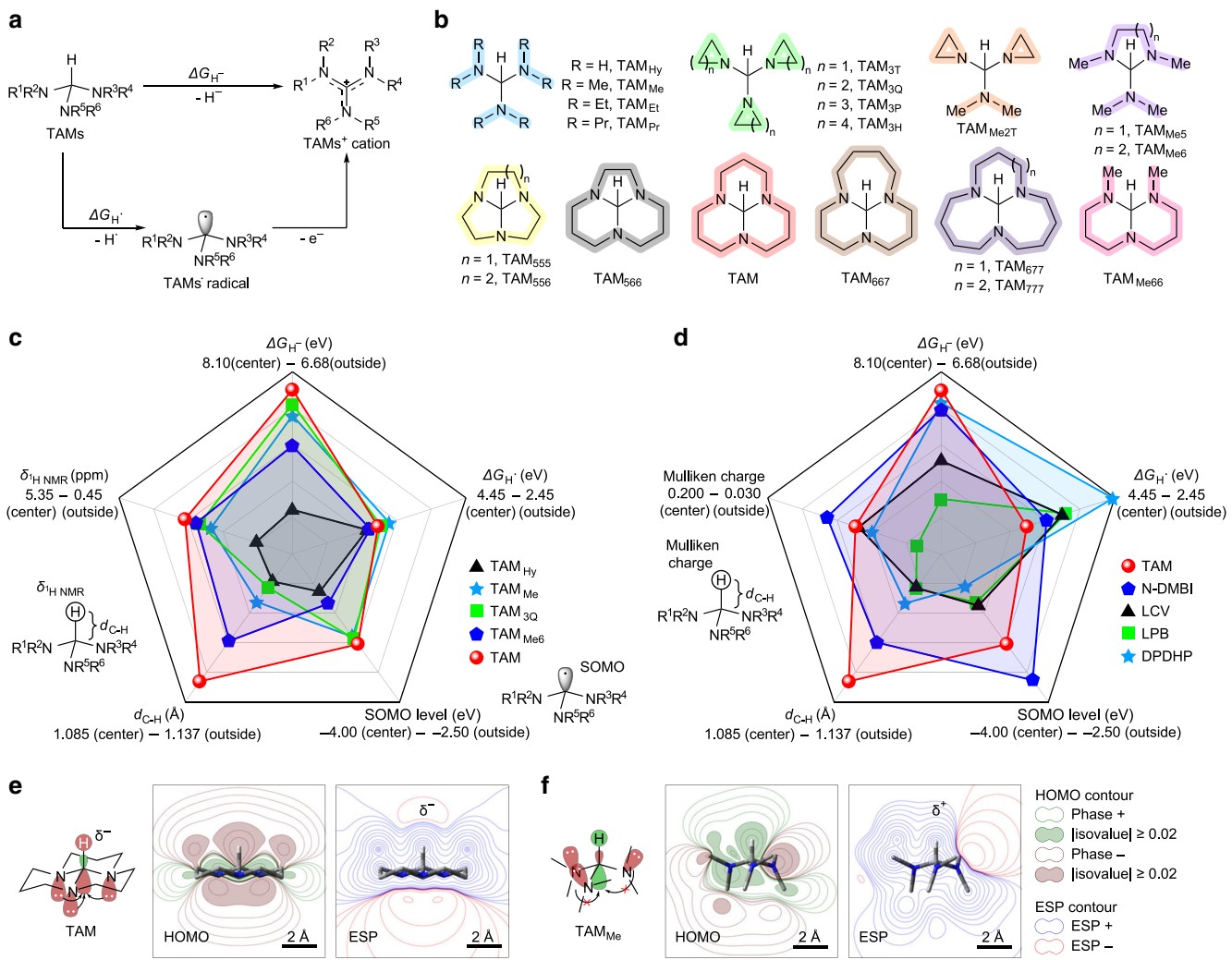

**Fig. 1 Dopant design and *n*-doping ability prediction. a** Design of triaminomethanes (TAMs) dopants with their guanidine-type cations. **b** Chemical structures of some designed TAMs. **c, d** Radar charms for predicting *n*-doping ability of TAMs (**c**) reported hydride dopants (**d**). Evaluating indexes includes density functional theory (DFT) calculated Gibbs free energy change in hydride-transfer half-reaction ($\Delta G_{H^-}$) and hydrogen-atom transfer half-reaction ($\Delta G_{H\bullet}$), singly occupied molecular orbital (SOMO) level of dopant radical, charge distribution (Mulliken charge or $^1$H-NMR chemical shift) of reactive hydrogen, and hydrogen-carbon bond length ($d_{C-H}$). **e, f** Secondary orbital interactions, highest occupied molecular orbital (HOMO) and electrostatic potential (ESP) contours of TAM and TAM$_{Me}$. DFT calculations performed under B3LYP/6-311 + G(d,p) level. TAM and other TAMs with 1,5,7-triazabicyclo[4.4.0]decane backbone (TAM$_{566}$, TAM$_{667}$, and TAM$_{Me66}$) show the highest predicted *n*-doping ability among TAMs.

Molecular dynamics (MD) simulations were performed to predict the interactions between the dopant cations and polymer side chains[27,44,45]. Fluorinated benzodifurandione-based poly(*p*-phenylene vinylene) (FBDPPV) anions are used in the MD simulation to study their different cation behaviors between TAM and N-DMBI (Fig. 2a). The geometric structures of FBDPPV anions and dopant cations were optimized with DFT calculations (Supplementary Fig. 4). The optimized structures were put into a supercell with periodic-boundary condition, where the polymer $\pi$–$\pi$ distances are set according to the grazing-incidence wide-angle X-ray scattering (GIWAXS) results (3.42 Å), and the polymer lamellar distances are lengthened to 50 Å (32.9 Å for GIWAXS results) to make the system became non-interdigitated (Fig. 2b)[45]. The initial positions of counterion TAMs$^+$ or N-DMBI$^+$ are kept in alkyl side-chain packing region with distances of 5.1 ~ 5.2 Å away from the polymer backbone (Fig. 2b and Supplementary Fig. 6). After the system reached equilibrium (Fig. 2c, d), most of TAM$^+$ cations stayed within the alkyl side-chain packing regions, away from the polymer-conjugated backbones, whereas a significant amount of N-DMBI$^+$ cations

were close to polymer-conjugated backbones. Statistical analysis of equilibrium-state counterion-backbone distances shows that TAM$^+$ cations pervasively move away from the backbone toward the middle of alkyl side chains, whereas N-DMBI$^+$ cations move toward the backbone or tails of alkyl side chains (Fig. 2e, f and Supplementary Fig. 7). Additional MD simulation with a different initial supercell size (lamellar distance = 42 Å, other conditions keep the same) of TAM/N-DMBI-doped FBDPPV shows consistent results (Supplementary Fig. 8), suggesting the simulation results do not depend on the supercell sizes. The different behaviors of the two cations could be explained by their different affinity properties: TAM$^+$ cations have similar polarizability property with alkyl side chains, while N-DMBI cations have similar polarizability property with polymer backbones (Supplementary Table 3, Supplementary Figs. 9 and 10, and Supplementary Note 2). Furthermore, TAM$^+$ cations can form tight packing structures with alkyl chains and have stronger affinities with alkyl chains than N-DMBI$^+$ cations (Supplementary Fig. 11). Thus, TAM$^+$ cations can be stabilized in alkyl side-chain regions, whereas the N-DMBI$^+$ cations are "crowded out" by the alkyl

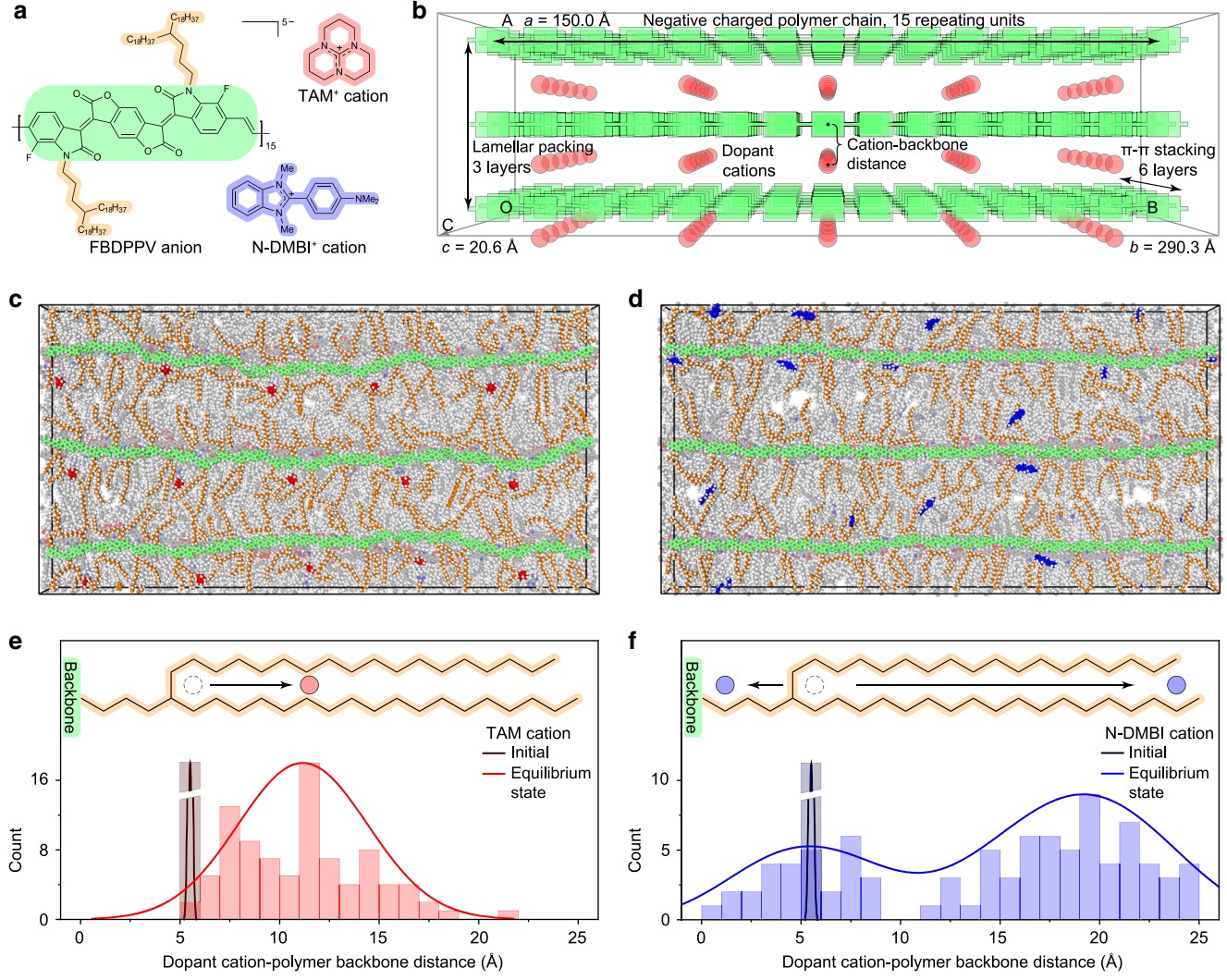

**Fig. 2 Cation–side chain interaction prediction. a** Chemical structures of FBDPPV anions, TAM$^+$, and N-DMBI$^+$ cations. **b** Schematic diagram of initial supercell for molecular dynamics which contains FBDPPV anions and TAM$^+$ (or N-DMBI$^+$) cations. Polymer alkyl chains are hidden for clearer vision. **c**, **d** Equilibrium configurations of n-doped FBDPPV by TAM (**c**) or N-DMBI (**d**). The middle (fourth of six) π–π stacking layer is highlighted for clearer vision. **e**, **f** Histogram of distances between dopant cations and polymer-conjugated backbones for TAM-doped FBDPPV (**e**) and N-DMBI-doped FBDPPV (**f**). Notice distance = 0 is polymer backbone and distance = 25 Å is alkyl chain tails. In all the above schematic diagrams, light green is for FBDPPV backbone, orange is for FBDPPV alkyl side chain, and red/blue is for dopant cation. TAM$^+$ cation has distinctly weaker interactions with polymer backbones, whereas N-DMBI$^+$ cation presents stronger interactions with polymer backbones.

chains to empty space, suggesting that N-DMBI doping could lead to severe phase seperations[11,22,24]. All these results demonstrate that TAM have better dopant-semiconductor miscibility than N-DMBI and TAM doping can form a "dopant-at-side-chain" microstructures.

**Synthesis, stability, and doping kinetics.** Although TAMs with 1,5,7-triazabicyclo[4.4.0]decane backbone (TAM, TAM$_{566}$, TAM$_{667}$, and TAM$_{Me66}$) are predicted to be strong hydride n-dopants, only TAM is synthesized in this work. TAM can be synthesized in a 10 g-scale through a one-pot reaction under air and at room temperature as shown in Fig. 3a. Pure TAM is a colorless crystal with high stability and good solubility (Supplementary Figs. 12 and 40). TAM is stable in air, under the sun light, in boiling water, and even in boiling concentrated hydrochloric acid (Fig. 3b and Supplementary Fig. 15). TAM is highly soluble in common organic solvents, including hexane, octane, toluene, acetone, dimethylsulfoxide, methanol, ethanol, chloroform, trichloroethylene (TCE), chlorobenzene, and 1,2-

dichlorobenzene (ODCB). In contrast, N-DMBI is neither as soluble nor as stable as TAM (Supplementary Figs. 12 and 13), limiting the applications of N-DMBI in certain solvents and conditions.

Although TAM has its extraordinary stability even in boiling water and acid, it can still efficiently transfer hydride to electrophile or efficiently n-doping organic semiconductors. TAM can transfer hydride to representative strong electrophile tritylium ions (Tr$^+$)[46] to generate triphenylmethane in high yield at room temperature in a few seconds (Fig. 3a, Supplementary Fig. 16, and Supplementary Note 4). For weaker electrophile such as tris(4-methoxyphenyl)methylium ((MeO)$_3$Tr$^+$), the hydride-transfer reaction of TAM is slower than that of N-DMBI (Fig. 3c and Supplementary Fig. 17). Reaction kinetics experiments show that for pseudo first-order hydride-transfer reactions, N-DMBI has a 284 times larger apparent rate constant ($k_{obs}$) compared with TAM (Supplementary Fig. 17 and Supplementary Note 5). DFT calculations on hydride-transfer transition states also shows that TAM has obviously higher activation energy ($\Delta G^{\ddagger}_{298K}$) but

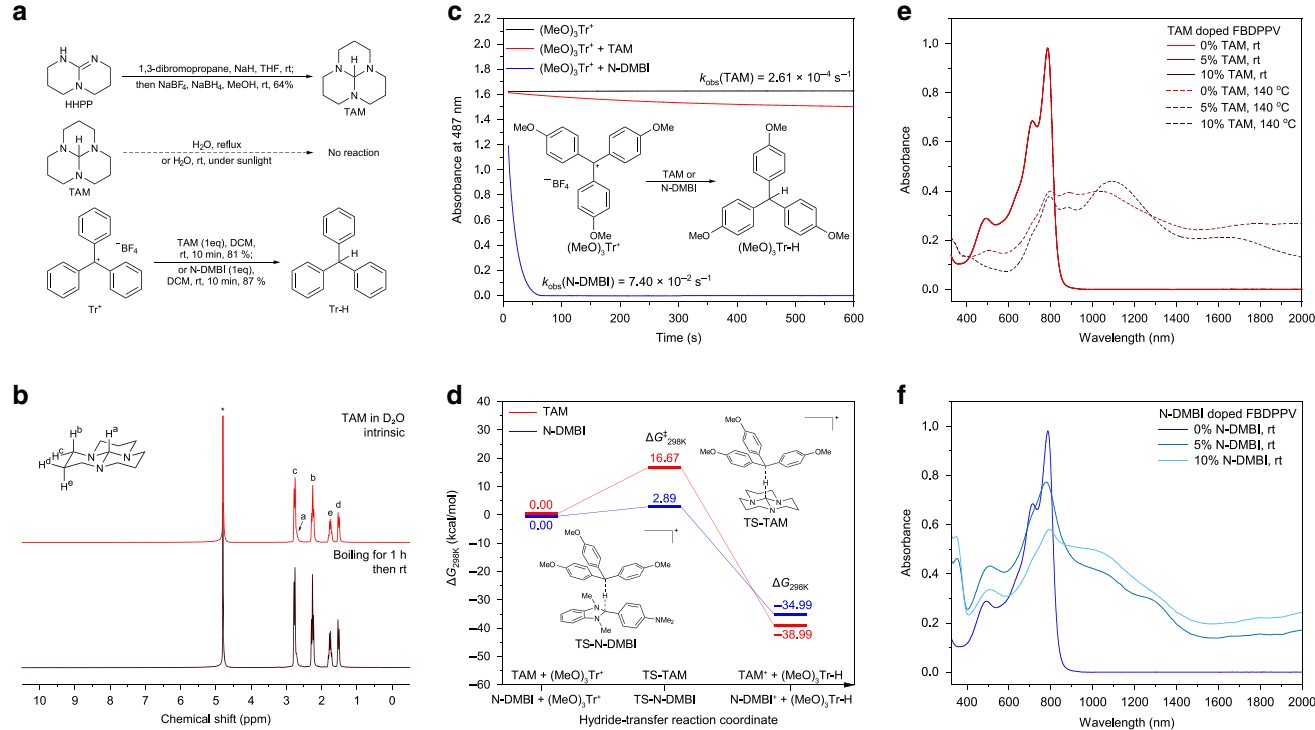

**Fig. 3 Synthesis, chemical property, and *n*-doping kinetics. a** Synthesis, chemical stability, and hydride-transfer ability of TAM. **b** Time-dependent $^1$H-nuclear magnetic resonance ($^1$H-NMR) spectra of TAM in boiling $H_2O$-$d_2$. **c** Time-dependent absorbance of hydride-transfer reaction between $(MeO)_3Tr^+$ and TAM (or N-DMBI). The initial concentrations are $2.38 \times 10^{-5}$ M for $(MeO)_3Tr^+$, $1.42 \times 10^{-4}$ M for both TAM and N-DMBI in anhydrous dichloromethane. **d** Density functional theory (DFT) calculated activation $\Delta G^{\ddagger}_{298K}$, reaction $\Delta G_{298K}$, and energy surface for hydride-transfer reaction between $(MeO)_3Tr^+$ and TAM (or N-DMBI). DFT calculation performed under $\omega$B97XD/6-311 + G(d,p)//B3LYP/6-31 G(d) level. **e, f** Absorption spectra of intrinsic and TAM (**e**)- or N-DMBI (**f**)-doped FBDPPV dilute 1,2-dichlorobenzene solution. The polymer concentration keeps 20 mg/L (in 1,2-dichlorobenzene) for each case, with different mass fraction of dopants after mixing for 5 min. TAM has ultra-high stability with strong *n*-doping ability at high temperature.

lower reaction energy change ($\Delta G_{298K}$) compared with N-DMBI (Fig. 3d), which is consistent with the reaction kinetic study. Hydride-transfer transition state calculations on other substituted tritylium ions and other typical electrophiles show that TAM has consistently higher $\Delta G^{\ddagger}_{298K}$ and lower $\Delta G_{298K}$ in hydride-transfer reactions (Supplementary Figs. 18 and 19, and Supplementary Note 6). These results demonstrates that TAM is a kinetically weaker but thermodynamically stronger hydride nucleophile compared with N-DMBI[46]. As both hydride-transfer and hydrogen-atom-transfer pathways have the same transition state[42], TAM can be predicted to have higher activation energies than N-DMBI in *n*-doping reactions.

Absorption spectra shows that N-DMBI can *n*-dope FBDPPV at room temperature, whereas TAM cannot (Fig. 3e, f, Supplementary Fig. 22, and Supplementary Note 10). However, after heating to 140 °C, TAM can efficiently accomplish *n*-doping of FBDPPV. Furthermore, TAM can only *n*-dope N2200 at high temperatures (Supplementary Figs. 20 and 21), similar to FBDPPV. This suggests that TAM doping needs a thermal activation process, because TAM is a kinetically weaker hydride donor. This implies that the thermally activated *n*-doping property is the key point to balance air/solution stability and doping ability of TAM. The thermally activated *n*-doping property also indicates that TAM-semiconductor blend solution can be kinetically stable, which is convenient for thick films deposition. The *n*-doping reactions of TAM are also characterized by ultraviolet photoelectron spectroscopy (UPS), and X-ray photoelectron spectroscopy (XPS) measurements (Supplementary Fig. 23). UPS measurement shows that the secondary electron cutoffs of both FBDPPV and N2200 shift obviously to higher

binding energies, indicating decreased work functions after the effective TAM *n*-doping (Supplementary Fig. 23a–d). N(1s) XPS measurement shows that quadrivalent nitrogen signals arise at 402.0 eV in TAM-doped FBDPPV and N2200, distinct from the lactam nitrogen (400.5 eV) in FBDPPV or imide nitrogen (400.9 eV) in N2200, indicating that TAM converts to guanidine cation after doping (Supplementary Fig. 23e–f). All these results suggest that TAM is an efficient *n*-dopant with weak kinetic activity but strong thermodynamic activity, and thus TAM is a thermally activated *n*-dopant.

**Thermoelectric performance with thick polymer films.** Thermoelectric devices are widely used in energy conversion and local temperature control. Compared with thermoelectric metal alloys, polymer-based materials are featured with low toxicity, low thermal conductivity, solution processability, and inherent flexibility[13]. Thermoelectric properties of polymers can also be evaluated in terms of the figure of merit ZT or power factor (PF):

$$ZT = \frac{S^2\sigma}{\kappa}T \qquad (1)$$

$$PF = S^2\sigma \qquad (2)$$

where $\sigma$ is electrical conductivity (S cm$^{-1}$), $S$ is Seebeck coefficient ($\mu$V K$^{-1}$), and $\kappa$ is thermal conductivity (W m$^{-1}$ K$^{-1}$) of a polymer. Geometry and thickness of thermolegs also play an important role in power conversion efficiency and power output maximum, as well as the figure of merit ZT and power factor of thermoelectric materials[47]. Compared with nanometer-thick thermoleg, micrometer-thick or even thicker thermoleg can

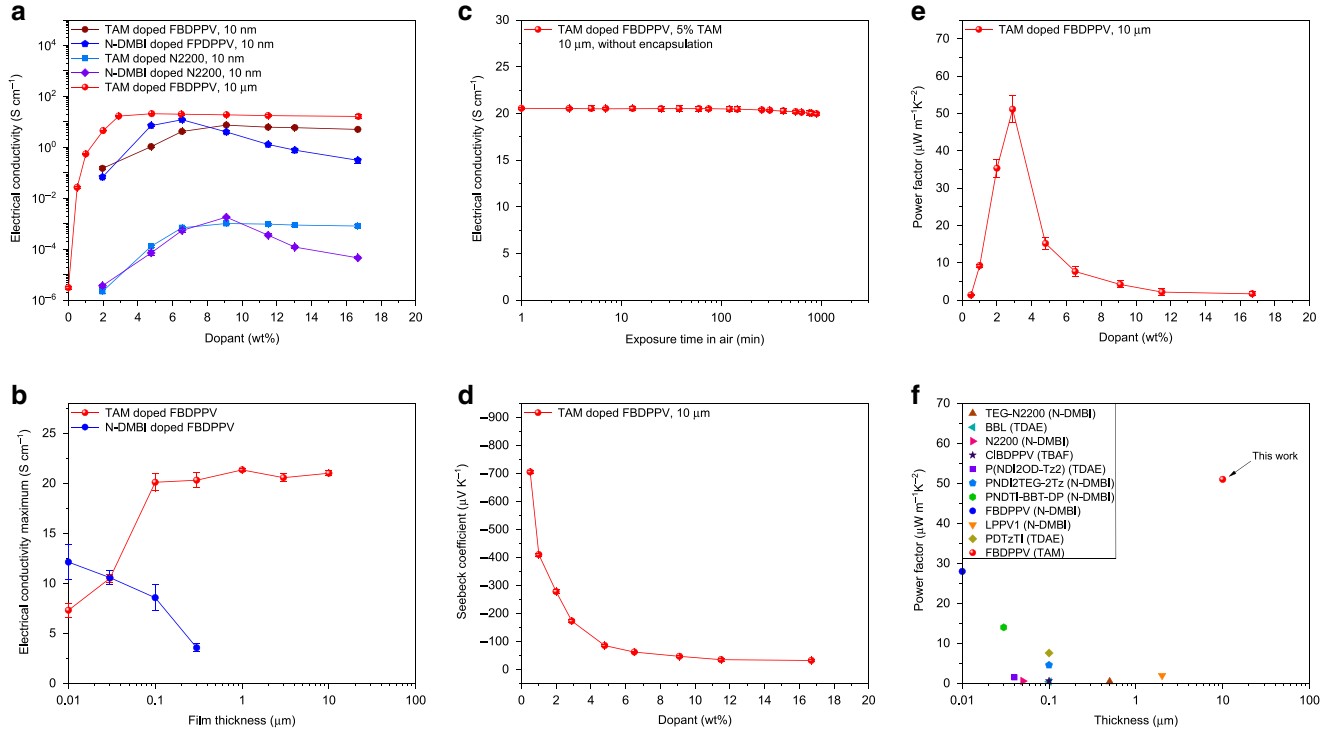

**Fig. 4 Thermoelectric performance of TAM-doped organic semiconductors. a** Dopant fraction-dependent electrical conductivity of TAM- and N-DMBI-doped polymers. **b** Thickness-dependent electrical conductivity maxima of TAM- and N-DMBI-doped FBDPPV. **c** Air stability of TAM-doped FBDPPV thick films without encapsulation. **d** Seebeck coefficients of TAM-doped FBDPPV in thick films. **e** Power factor of TAM-doped FBDPPV in thick films. **f** Thermoelectric performance comparison of TAM-doped FBDPPV with literature results (refs. [11,24,26–30,53–58]). Error bars indicate the SD of ten experimental replicates. TAM-doped polymers present ultra-high thermoelectric performance in high thickness.

hold larger effective temperature differences or hold more effective heat flow thus to achieve higher $\eta_{max}$ or $P_{max}$ in thermoelectric generators (see detailed calculation and discussions in Supplementary Figs. 25–28 and Supplementary Note 7)[48].

Various mass fractions of TAM or N-DMBI were used to $n$-dope FBDPPV and N2200 to optimize polymer electrical conductivity (Fig. 4a). The doped polymer thin films (10 nm) were fabricated through spin-casting from ODCB solutions of blended dopant and polymer, and following thermal annealing (140 °C). The electrical conductivity of TAM-doped FBDPPV increases rapidly with dopant fraction, reaches a maximum of 7.21 S cm$^{-1}$ at 9% of dopant, and decreases very slowly when more dopants were added. In contrast, for N-DMBI-doped FBDPPV, its electrical conductivity reaches a maximum of 12 S cm$^{-1}$ at 7% of dopant ratio, but then decreases rapidly as more dopants were added. TAM- or N-DMBI-doped N2200 showed consistent trends as doped FBDPPV. These results can be explained with the different solid-state microstructures caused by different dopants. This will be discussed later.

The high solution stability and thermally activated doping ability make TAM easily to fabricate doped polymer films with thicknesses of $10^{-2} \sim 10$ μm through simple spin-casting or drop-casting, followed by thermal annealing (140 °C). On the contrary, the instability of N-DMBI in TCE or chloroform makes the fabrication of N-DMBI-doped FBDPPV films more restricted: they can only be processed in ODCB and we could never get films thicker than 0.3 μm. The optimized electrical conductivities of TAM- and N-DMBI-doped FBDPPV in various thickness are plotted in Fig. 4b. The maximum electrical conductivity of N-DMBI-doped FBDPPV decreases with increasing of film thickness, whereas that of the TAM-doped FBDPPV films increases with the increasing of film thickness, finally reaching 20 S cm$^{-1}$ at

thicknesses above 0.1 μm. In 10 μm-thick films, TAM-doped FBDPPV reaches the maximum electrical conductivity of 21.0 ± 0.24 S cm$^{-1}$ with 5% of TAM (FBDPPV repeating unit: TAM = 2.1 mol/mol), which is among the highest conductivity of $n$-type polymers. This result indicates that TAM has strong thermodynamic $n$-doping ability, consistent with the above theoretical and experimental studies. The electrical conductivity of TAM-doped FBDPPV thick films does not decrease significantly at higher dopant fractions, similar to its thin films (Fig. 4a). Air stability in one of the key issues in high-performance $n$-type thermoelectric materials[49,50]. The TAM-doped FBDPPV thick films are stable in air at room temperature. The electrical conductivity of the unsealed thick film does not decrease obviously when exposed to the air ($R_H$ = 50 ~ 60 %), suggesting this thick film is compact enough to prevent bulk oxidative de-doping at room temperature (Fig. 4c). On the contrary, TAM-doped FBDPPV thin films are unstable in air due to the lack of the above self-encapsulation (Supplementary Fig. 29). Moreover, TAM-doped BDPPV thick films are also stable during long-time thermal annealing in nitrogen (but unstable during thermal annealing in air, see Supplementary Fig. 29 and Supplementary Note 8), indicating that TAM doping is irreversible. TAM-doped FBDPPV thick films have negative Seebeck coefficients of −705 ~ −32 μV K$^{-1}$, which decrease as increasing of the dopant fraction (Fig. 4d and Supplementary Figs. 31 and 32). In addition, the maximum power factor of TAM-doped FBDPPV thick films reaches 51 ± 3.6 μW m$^{-1}$ K$^{-2}$ with 3% of TAM (FBDPPV repeating unit: TAM = 3.6 mol/mol), which is also among the highest in $n$-type polymers (Fig. 5e, f).

In addition, TAM can $n$-dope small-molecule organic semiconductors. TAM-doped PC$_{61}$BM shows a maximum electrical conductivity over 0.04 S cm$^{-1}$ (Supplementary Fig. 35).

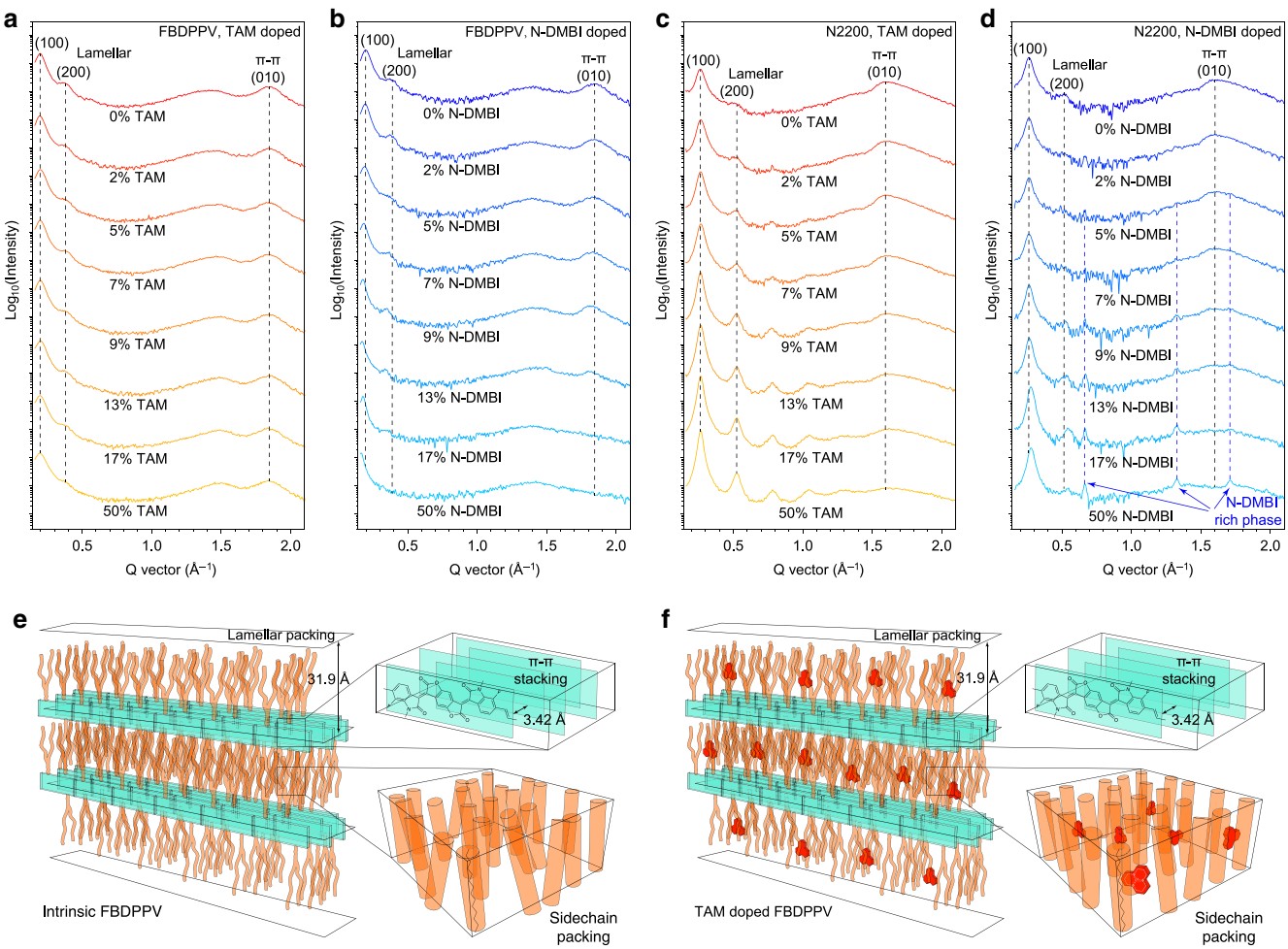

**Fig. 5 Solid-state microstructures. a–d** Out-of-plane grazing-incidence wide-angle X-ray scattering (GIWAXS) analysis of intrinsic and doped polymers: TAM-doped FBDPPV (**a**), N-DMBI-doped FBDPPV (**b**), TAM-doped N2200 (**c**), and N-DMBI-doped N2200 (**d**). **e, f** Schematics of the proposed molecular packing models in the intrinsic (**e**) and TAM-doped (**f**) FBDPPV, where light green is for FBDPPV backbone, orange for FBDPPV alkyl side chain, and red for TAM$^+$ cation. TAM doping will not affect polymer $\pi$–$\pi$ stacking and lamellar packing, whereas N-DMBI doping can affect both polymer $\pi$–$\pi$ stacking and lamellar packing. TAM is a highly miscible dopant and TAM doping presents "dopant-at-side-chain" microstructures.

Besides, TAM can $n$-dope organic semiconductors by vapor. TAM vapor-doped FBDPPV thin film shows a maximum electrical conductivity of over 6 S cm$^{-1}$, which is comparable to the solution-process doped thin films (Supplementary Fig. 30). Thus, TAM can be used in $n$-doping organic semiconductors through various processing methods.

**Solid-state microstructures.** GIWAXS was performed to study the microstructures of semiconductors doped by TAM and N-DMBI. Intrinsic FBDPPV shows lamellar packing ($h$00) with distance of 31.9 Å, and $\pi$–$\pi$ stacking (010) with distance of 3.42 Å (Fig. 5a, b, e and Supplementary Fig. 42). After TAM doping, the lamellar packing distance and $\pi$–$\pi$ stacking distance of FBDPPV remain almost the same (Fig. 5a and Supplementary Figs. 36 and 42). For N-DMBI-doped FBDPPV, its lamellar packing distance gradually increases to 38.2 Å and its $\pi$–$\pi$ stacking distance increases to 3.47 Å at 13% dopant fraction. The $\pi$–$\pi$ stacking diffraction vanishes at 17% or higher dopant fraction (Fig. 5b and Supplementary Figs. 37 and 42). These results indicate that the N-DMBI$^+$ cations can destroy the polymer packing, which is consistent with the MD simulation that a portion of N-DMBI$^+$ cations locate in lamellar interspace and the others locate in the $\pi$–$\pi$ stacking region. In contrast, the good miscibility between

TAM$^+$ and alkyl side chains leads to the "dopant-at-side-chain" microstructure, where TAM$^+$ most probably locate in the alkyl side-chain packing region (Fig. 5f and Supplementary Fig. 11). The "dopant-at-side-chain" microstructure keeps the lamellar packing distance and $\pi$–$\pi$ stacking distance constant. In addition, the homogeneous doping properties are also observed in the TAM-doped thick films (Supplementary Figs. 44 and 45). The microstructure uniformity could explain the high electrical conductivities of TAM-doped FBDPPV in various thicknesses. In contrast, the lower electrical conductivity in the thick films of N-DMBI-doped FBDPPV might be probably caused by its inhomogeneous microstructure. More importantly, TAM doping will not reduce but slightly enhance electron mobility of FBDPPV and this effect is more pronounced than N-DMBI doping (Supplementary Fig. 24). TAM-doped FBDPPV demonstrates weaker temperature dependence of both electrical conductivity and See-beck coefficient (Supplementary Figs. 33 and 34, and Supplementary Note 9), indicating that TAM-doped FBDPPV has lower molecular $\pi$–$\pi$ stacking disorder than N-DMBI-doped FBDPPV (Supplementary Fig. 42 and Supplementary Note 11).

Furthermore, TAM doping can also lead to homogeneous microstructures in doped N2200 (Fig. 5c and Supplementary Figs. 38 and 43), whereas N-DMBI doping leads to phase separation in doped N2200 (Fig. 5d and Supplementary Figs. 39

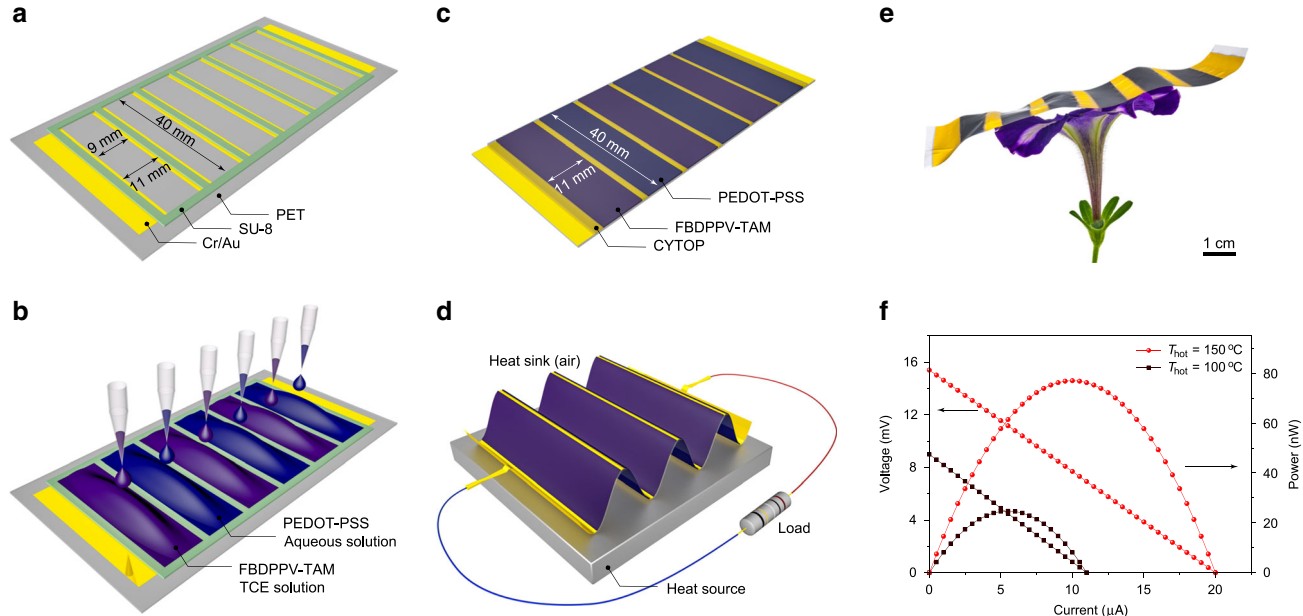

**Fig. 6 Flexible all-polymer solution-processed thermoelectric generator. a–d** Fabrication and testing: Au electrodes and SU-8 isolation columns patterning (**a**); PEDOT-PSS, FBDPPV-TAM printing (**b**); cutting and insulating encapsulation (**c**); power output measurement in air (**d**). **e** Photograph of flexible thermoelectric generator on a petunia. **f** Output voltage and power of the generator at different heat source temperature (air temperature is 25 °C).

and 43). All these results indicate that TAM is a highly miscible dopant with negligible negative effects on polymer packing and charge transporting.

**All-polymer thermoelectric generator**. The solution processability, high stability, and good thermoelectric performance of TAM-doped semiconductors allow us to fabricate all-polymer thermoelectric generator. PEDOT-PSS (PH 1000), which is commercially available and solution processable, is employed as the *p*-leg material and the FBDPPV-TAM is employed as the *n*-leg material. The thermoelectric generator including three *p–n* modules was fabricated on a flexible polyethylene terephthalate (PET) substrate. First, gold electrodes and SU-8 isolation columns were patterned by photolithography (Fig. 6a). Then, PEDOT-PSS *p*-legs and FBDPPV-TAM *n*-legs were printed through drop-casting (Fig. 6b). Finally, SU-8 isolation columns were removed and the device was encapsulated by a perfluorinated cyclic transparent optical polymer (CYTOP) layer (Fig. 6c). The thermoelectric generator is light and flexible (Fig. 6e). This thermoelectric generator was corrugated on a hot plate and the power output was tested in air (Fig. 6d, Supplementary Figs. 52 and 53, and Supplementary Movie 1). The generator shows a maximum power output of 0.18 nW at $T_{hot} = 30$ °C (with temperature gradient $\Delta T = 2.3$ K) and the maximum power output reaches 25 nW and 77 nW after $T_{hot}$ increasing to 100 °C and 150 °C ($\Delta T = 27.8$ K and 46.5 K, respectively), even though the generator only has an air heat sink and its thermoleg geometry was not fully optimized (Fig. 6f and Supplementary Fig. 53a–f). The thermoelectric generator also shows stable power output during bending its thermolegs (bending radius = 1 cm, Supplementary Fig. 53g–h). Compared with some polymer thermoelectric generators with only PEDOT-PSS *p*-leg and metal connection lines, this complementary all-polymer generator demonstrates obviously enhanced power output (Supplementary Fig. 53i).

## Discussion

We have developed a high-performance organic *n*-dopant TAM through the assistance of DFT calculations and MD simulations.

TAM is also first applied as an *n*-dopant and TAM presents several unique features: extraordinarily high stability and good solubility; thermally activated *n*-doping property with kinetically weak but thermodynamically strong doping ability; outstanding miscibility with polymers and "dopant-at-side-chain" microstructures with little impacts on polymer π–π stacking and charge transport to exhibit excellent *n*-doping efficiency. As a result, TAM-doped polymer thick films realizes currently the highest electron conductivity and power factor among all solution processable *n*-type thermoelectric polymers. Due to its convenient synthesis and excellent processability, the utility of TAM solves the long-standing challenge of the low-doping efficiency and poor miscibility of polymer/dopant systems and enables high-performance *n*-type organic thermoelectrics. We believe that TAM could also be used as *n*-dopant for other types of semiconductors to adjust their electron concentration and improve their semiconductor device performance.

## Methods

**Density functional theory calculations**. Geometry optimizations and n-doping ability evaluating indexes calculation on TAMs and other hydrides were performed under B3LYP/6-311 + G(d,p) level. For dopant radicals, uB3LYP/6-311 + G(d,p) was employed. For isotropic polarizability of dopant cations, B3LYP/6-311 + G(d, p) were used. Hydride-transfer transition state calculations were performed under ωB97XD/6-311 + G(d,p)//B3LYP/6-31 G(d) level. For absorption spectra calculations, geometry optimization on FBDPPV trimer, trimer anion and trimer dianion were performed under B3LYP/6-31 G(d) level. Time-dependent DFT calculations on vertical excitation energy were performed under B3LYP/6-31G(d) level. For trimer anion, uB3LYP/6-31G(d) were employed.

**Molecular dynamics simulation**. MD simulation was performed with Materials Studio package1 according to the previous procedure reported by Sirringhaus and co-workers[45]. First, geometry structures of FBDPPV trimer anion, TAM, and N-DMBI cations were optimized by DFT calculation (Supplementary Figs. 2 and 4) and charges of each atom were assigned by Gasteiger method[44]. Second, FBDPPV polymer crystal structure was constructed from the optimized trimer structures. Third, the orthogonal supercells with periodic-boundary condition containing three lamellar layers of six π–π stacked polymer chains (5 trimers, 15 repeating units) were built ($a$ = 150.0 Å, $b$ = 290.3 Å, $c$ = 20.6 Å). Then dopant cations were appropriately added into the polymer side-chain stacking interspace until net charge of the system was zero (Supplementary Fig. 6). The molar ratio of FBDPPV repeating unit and dopant cation is 3 : 1. The whole system contained 270 polymer repeating units and 90 dopant cations, which reached

molecular weight of about $5 \times 10^5$ Daltons. Lastly, an MD simulation of 50 ps in the NPT ensemble ($P = 1 \times 10^5$ Pa, $T = 500$ K) was followed by an MD of 200 ps at room temperature in the NPT ensemble ($P = 1 \times 10^5$ Pa, $T = 298$ K). The equilibrium states were extracted to analyze the dopant distribution in polymer matrix and distances between each dopant cation and its nearest polymer backbone were measured (Supplementary Fig. 5) for statistical analysis. In all MD calculations, the DREIDING force field 3, Nosé–Hoover thermostat, and time step of 1 fs were used[51].

**Materials**. N-DMBI, N2200 ($M_n = 75.2$ kDa, PDI = 2.47) and FBDPPV ($M_n = 153$ kDa, PDI = 2.39) were synthesized following the procedures from previous literatures[11,52,53]. PEDOT-PSS (PH 1000) was purchased from Sigma-Aldrich (https://www.sigmaaldrich.com). One-pot 10 g-scale synthesis procedure of TAM is showed below. To a 500 mL two-neck bottle equipped with calcium chloride drying tube, 5.75 g of sodium hydride (60% in naphtha, 144 mmol) was added and washed by hexane. A solution of HHPP (20.0 g, 144 mmol) in 300 mL of anhydrous tetrahydrofuran was added to the washed sodium hydride and 29.0 g of 1,3-dibromopropane (144 mmol) was added dropwise. Then the mixture was stirred for 24 h in room temperature. During the stirring, a large amount of white precipitate was produced. The white precipitate was collected through vacuum filtration, washed with 200 mL of anhydrous ether, and then added to sodium tetrafluoroborate saturated aqueous solution (63.10 g of NaBF$_4$, 575 mmol). The resulting aqueous solution was extracted with dichloromethane ($5 \times 100$ mL). The organic phase was combined and then dried with 10 g of anhydrous sodium sulfate and all solvents was removed by vacuum distillation. Cooled methanol (500 mL) containing 0.2 mL of water was added to dissolve the resulting intermediate product and 27.2 g of sodium borohydride (718 mmol) was added in several batches. The reaction temperature was kept below 25 °C by cold water bath. After stirring for 12 h, all solvents were removed by vacuum distillation and 250 mL of water was added. The aqueous solution was extracted by dichloromethane ($5 \times 100$ mL) and organic phase was combined and then dried with 10 g of anhydrous sodium sulfate. Afterwards, solvents were removed by vacuum distillation to give crude TAM. The crude TAM was dissolved in 120 mL of anhydrous hexane, dried with 10 g of anhydrous sodium sulfate, and filtrated to a 250 mL flask. Lastly, the hexane that may contain trace water was removed in vacuum to give pure TAM as colorless transparent crystals (16.6 g, 64% yield). $^1$H-NMR (CDCl$_3$, 400 MHz, p.p.m.): $\delta$ 2.80 (m, 6H), 2.31 (s, 1H), 2.14 (m, 6H), 2.05 (m, 3H), 1.43 (m, 3H). $^{13}$C NMR (CDCl$_3$, 101 MHz, p.p.m.): $\delta$ 100.0, 53.8, 24.1. ESI HRMS calcd. for (M + H)$^+$: 182.1652; Found: 182.1654.

**Thermoelectric performances**. Comparison of thermoelectric power factor of reported *n*-type polymers and TAM-doped FBDPPV are showed in Fig. 4f[11,24,26–30,54–58].

**All-polymer thermoelectric generator**. The 12.5 μm-thick PET substrate was cleaned with isopropanol and dried with nitrogen, then stretched over a flat stainless steel framework. The 100 nm parallel gold electrodes along with 2 nm of chromium adhesion layer was vacuum evaporated to the substrate and then patterned by photolithography and wet chemical etching. Each electrode was patterned to 4 mm wide and 40 mm long. The gap between gold electrodes was 9 mm. The 50 μm-thick SU-8 isolation columns were patterned by lithography to give carves for thermolegs patterning and they are 11 mm wide and 40 mm long. Then the processed substrate was cleaned by oxygen plasma for 10 min to improve hydrophilicity. PEDOT-PSS was printed by drop-casting, annealed at 130 °C, washed by methanol, and then dried in nitrogen to give three *p*-legs with a thickness of 10 μm. FBDPPV-TAM was printed by drop-casting, then annealed at 140 °C under nitrogen atmosphere to give three *n*-legs with thickness of 10 μm. SU-8 isolation columns were then removed by cutting and peeling off. CYTOP insulating encapsulation layer with thickness of 500 nm was spin-casted and annealed at 100 °C to finish the fabrication of thermoelectric generator. Comparison of maximum power output (per number of thermos modules) of some polymer (including coordination polymer) thermoelectric generators are showed in Supplementary Fig. 53i[59–63].

## Data availability
The data that support the findings of this study are available from the corresponding authors upon reasonable request.

## Code availability
The code that support the findings of this study are available from the corresponding authors upon reasonable request.

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

## Acknowledgements

This work was supported by Key-Area Research and Development Program of Guangdong Province (2019B010934001), National Key R&D Program of China (2017YFA0204701), and National Natural Science Foundation of China (21790360, 21722201, and 21420102005). We thank beamline BL14B1 (Shanghai Synchrotron Radiation Facility) for providing beam time. We thank Dr. Yi Wang from College of Chemistry and Molecular Engineering, Peking University, for help discussing DFT calculation results. We thank Dr. Han-Yan Wu from Academy for Advanced Interdisciplinary Studies, Peking University, for help discussing MD simulation results. We thank Dr. Matteo Massetti from Department of Science and Technology, Linköping University, for help taking SEM images.

## Author contributions

C.-Y.Y. designed TAM. C.-Y.Y. performed the DFT and TD-DFT calculations. Z.-F.Y. performed the MD simulations. C.-Y.Y. and J.W. synthesized TAM, N-DMBI, and FBDPPV. Y.-F.D. and F.-D.Z. synthesized N2200. Y.-F.D. performed the dopant stability experiments. Y.-F.D. and C.-X.H. performed hydride-transfer experiments. C.-Y.Y. and Y.-F.D. performed the doping ability experiments. C.-Y.Y., D.H., Y.-F.D., and H.-I.U. performed the thermoelectric property measurements. C.-Y.Y., Y.L., Y.-F.D., and J.-H.D. performed the GIWAXS measurements. J.W. performed the AFM measurements. C.-Y.Y. and Y.-F.D. performed the fabrication and measurement of thermoelectric generators. C.-Y.Y., C.-A.D., D.Z., J.-Y.W., T.L., and J.P. co-wrote the paper. All the authors discussed the results and commented on the manuscript.

## Competing interests

The authors declare the following competing interest: a pending patent application (CN 201910322353.6) entitled "Application of triaminomethane derivatives as *n*-dopants in semiconductor materials," which Peking University is the patent applicant and J.P., C.-Y.Y., J.-Y.W., T.L., Y.L., and Y.-F.D. are co-inventors. The contents of this patent application are based on the results described in the present study.
