## [Peer Review File · Nature Communications]

Reviewers' Comments:

Reviewer #1:

Remarks to the Author:

This manuscript reports interesting new n-dopant structures and rationales for their design. Rather than make a direct recommendation of whether the manuscript would be most appropriate for Nature Communications or another journal, the comments below illustrate some points of novelty and limits to the novelty. The paper on the whole merits publication subject to addressing the issues discussed below, and would be a valuable contribution to the literature.

The amine donors appear novel and insightful. Their analysis according to both kinetic barriers and thermodynamic driving forces for n-doping is also useful and novel. There is extensive computational data about structures considered before experimentation and characterization data for samples actually made.

It is not clear from what larger library of possible structures the ones in the present study were selected by the calculations. For example, were ethyl and propyl substituents screened, or other combinations of rings and methyl groups? Were the major structures conceived of by intuition and then evaluated, or were the structures generated by the computation?

The facile synthesis and high stability of the TAM-H compound itself are attractive features.

The comparison of hydride transfer activation energies and driving forces between TAM-H and N-DMBI is interesting. However, the high activation energy for TAM-H hydride transfer may not be desirable because a somewhat harsh heating step is required to complete the doping process. Also, the doping is not necessarily via hydride transfer; it may be from homolytic C-H bond cleavage, so the hydride transfer thermodynamics may not be highly relevant. The thermodynamics of doping via hydride or hydrogen atom would be equivalently irreversible for any of the triaminomethanes if H₂ gas were generated or if new, stronger C-H bonds were formed elsewhere in the polymer.

Thermodynamic parameters are favorable, especially for n-type thin films. However, more insight is needed about why the thick films of this work behave better than in previous works; is the morphology denser, or are the chemical structures inherently more difficult to dedope with oxygen? Is the mobility significantly higher in the doped form compared to other n-doped polymers, or is there some favorable density of states distribution that allows relatively high Seebeck coefficient along with the high conductivity? Figure S21 suggests slightly enhanced mobility from doping, but it is not clear whether this effect is more pronounced than with the N-DMBI dopant.

The absolute power factor seems about twice as high as other doped polymer systems, based on the authoritative review by one of the coauthors (Zhu) and as stated in the present manuscript. The stability of the electrical conductivity looks promising, but other n-type conducting polymers, such as nickel thiol coordination polymers (Wolfe et al., *Adv. Electron. Mater.* 2019, 5, 1900066), have been stabilized on similar or greater time scales.

Making a thermoelectric generator is a good demonstration. There is no comparison with other reported thermoelectric generators by which to judge the power output magnitude.

Some minor points:

The ZT of 0.42 for PEDOT is overstated because of the geometries of the electronic and thermal conductivity measurements not being the same, and the possible enhancement from ionic conductivity. See Liu et al., *Macromolecules* 2015, 48, 3, 585-591.

Caption to Figure 2, bottom of page 4: while the TAM⁺ cation may have less interaction with conjugated polymer chains than N-DMBI does, it is not likely to have a "strong" interaction with nonpolar/nonfunctional alkyl chains.

Reviewer #2:

Remarks to the Author:

The manuscript by Yang, et al. reports a new n-type dopant based on a triaminomethane derivative. The manuscript is thorough and systematically goes through the design of the dopant family, including analysis of a number of calculated parameters relevant to their doping behavior, and then reports the synthesis and doping behavior of one of the predicted high-performing derivatives. The new dopant, TAM, is shown to have high stability and is used to dope FBDPPV and achieve high n-type thermoelectric power factors, including in much thicker films than are accessible with the dopant N-DMBI. The manuscript provides a nice comparison between TAM and N-DMBI throughout. The development of this new n-type dopant, which may potentially lead to a larger family of n-type dopants, is significant and the work is detailed and thorough. It would be helpful if the authors can address the following concerns:

In Figure 1c, it is hard to distinguish some of the colors, for example TAM555 and TAM. I recommend using the same color scheme, but using differing symbols for each TAM derivative. The symbols should be placed to indicate the intersection with each axis. This would allow the derivatives to be more easily identified.

In the radar charms it is not initially clear whether the values (e.g., 5.5 to 1.5) are listed in

order from the outside to the center or vice versa. Clearly indicating whether the first value listed is the center or edge of the axis would be helpful.

The end of the Figure 1 caption says "TAM and TAM556 show the highest n-doping ability among TAMs." I think that TAM556 should be TAM566 and this should say the highest predicted n-doping ability. The statement as written led me to expect that these derivatives would all be experimentally synthesized and characterized, which is not the case. Only TAM is reported.

Why are the lamellar stacking distances lengthened from the observed 32.9 Å to 50 Å for the MD simulations? Lengthening this spacing introduces significantly more free volume which is likely going to lead to significant differences in the MD simulations as compared to if the experimentally observed lamellar stacking distances were used. This may be one of the reasons that the backbones are less planar in the TAM containing system than the N-DMBI system. This increased disorder in the conjugated backbone observed in the simulation for TAM is not apparent in the experimental GIWAXS measurements.

At the end of page 5 the text states: "the hydride-transfer reaction of TAM is slower than that of N-DMBI (Fig. 5d and Fig. 5g)." This text is not referring to the correct figures. Additional incorrect figure reference appears later in that same paragraph at the top of page 6.

The statement "UPS measurement shows that the secondary electron cutoffs of both FBDPPV and N2200 shift obviously to lower kinetic energies" is odd, as the secondary electron cutoff is looking at the electrons with near 0 kinetic energy. These kinetic energies are not changing. The authors likely mean that the secondary electron cutoff shifts to higher binding energies, indicating a decreased work function.

From the ultraviolet photoelectron spectra in Figure S20 the HOMO onset of the polymers cannot be seen due to the large y-scale used. A zoomed in view of the HOMO onset region should be included.

On page 7 the authors say "as merit ZT". This should say "as the figure of merit ZT". The authors should define both ZT and the power factor.

Plot 4c should be plotted with electrical conductivity on a linear scale to allow for changes to be more easily observed.

The discussion of the actual thermoelectric module is not too informative. Neither the figure nor discussion contain any dimensions and no measurement or even an estimate of the temperature gradient across the device is presented.

It would be helpful if the authors clearly stated early on in the paper that only TAM was synthesized. After reading through the theory section I was expecting multiple derivatives to be made and reported. Reporting only TAM is enough, but identifying this early on would be helpful to better direct expectations.

Reviewer #3:

Remarks to the Author:

The authors report on the synthesis of a new n-type dopant named TAM (based on triaminomethane) with high stability and strong thermally activated doping mechanism. TAM was then integrated into operational thermoelectric materials as a dopant, enabling creation of a thick (up to 10 μm) films with high output voltage and current.

The idea is to obtain the n-type dopant without all the conventional issues like stability and miscibility, which typically affect organic thermoelectric systems, and further use it for n-type doping.

The topic is highly interesting and deals with the complex and advanced approach of dopant formation, involving theoretical DFT-based calculations, in parallel to experiments. The target is to investigate the influence of the dopant in lamellar and π - π stacking of the targeted for doping polymers. The field of direct transformation of heat into electricity through the thermoelectric effect is fast evolving and attracts ever growing interest.

One of the main weaknesses in the field is the lack of n-type materials and thus the research presented in the manuscript is very timely and interesting. It can be of interest for the broad audience of Nature Communication, not only involved in the field of thermoelectric, but also interested in the field of organic electronics.

While the idea reported by the authors is original and appealing, there are some statements that should be addressed. A detailed analysis on the weaknesses of the manuscript is reported below.

On the basis of the observations below, I would suggest the authors to modify the manuscript according, before reconsidering it for publication.

Overall, I would suggest major revisions before accepting the manuscript for publication on Nature Communications.

Comments to the paper:

1. The authors state that TAM was discovered. They should clarify: was TAM firstly synthesized or firstly applied as a dopant?
2. Please consider the following publication with much higher values of power factor "ACS Appl. Mater. Interfaces 2019, 11, 3400–3406"
3. What parameters were used to calculate the DFT. Spin-polarized calculation? Spin-orbital coupling considered? +U parameter? Hybrid pseudopotentials? The authors should clarify this point.

4. The author should consider to split or enlarge Fig 3. In its present form, it is too difficult to read.
5. "Absorption spectra shows that N-DMBI can n-dope FBDPPV at room temperature," How exactly authors estimate it from absorption spectra? Which peak refers to what energy state and how electrons are connected? This aspect is not clear.
6. "The doped polymer thin films (10 nm) were fabricated through spin-casting from ODCB" Can authors show the SEM or TEM images exhibiting the thickness of the doped polymer?
7. "TAM-doped FBDPPV thick films are stable in air." is that statement valid at elevated temperatures? With increase of T, the TAM doped films are supposed to exhibit higher S, since TAM is thermally activated, thus changing the chemistry of TAM layer on top of the FBDPPV. Please clarify.
8. Output voltage should be converted from mW (which is not at all a voltage) to mV for the better comparison with other research results.
9. If authors emphasize the flexibility of the generators then the appropriate test should be conducted, i.e., the influence of folding times on TE properties. Additionally, the folding of the "legs" is shown in a way where the actual p- and n-type materials are not changing their shape whereas the substrate is bending.
10. Output parameters (voltage, current, power) are all acquired at T differences from 100 to 150 °C which are too far from differences that can be reached in real life. The authors should provide the same parameters at difference of 30 °C which is more realistic, or explain why this is not possible.

For Reviewer #1:

This manuscript reports interesting new n-dopant structures and rationales for their design. Rather than make a direct recommendation of whether the manuscript would be most appropriate for Nature Communications or another journal, the comments below illustrate some points of novelty and limits to the novelty. The paper on the whole merits publication subject to addressing the issues discussed below, and would be a valuable contribution to the literature.

The amine donors appear novel and insightful. Their analysis according to both kinetic barriers and thermodynamic driving forces for n-doping is also useful and novel. There is extensive computational data about structures considered before experimentation and characterization data for samples actually made.

Q1:

“It is not clear from what larger library of possible structures the ones in the present study were selected by the calculations. For example, were ethyl and propyl substituents screened, or other combinations of rings and methyl groups? Were the major structures conceived of by intuition and then evaluated, or were the structures generated by the computation?”

Our Response:

As the reviewer mentioned, the major structures of TAMs were conceived of by intuitive design and then evaluated. Here is the design and evaluation process:

- 1) We choose triaminomethane building block, because it can form a stable guanidine cation after reduction. This comes from our long-time molecular design intuitive and literature research.
- 2) We introduce alkyls to nitrogen atoms to enhance the electron-donating property of TAMs;
- 3) We investigate the effect of methyl and longer alkyl substitutions and evaluate the redox properties and stabilities of the compound by calculation;
- 4) Reducing steric hindrance by inspecting cyclic/fused cyclic alkyl substitutions;
- 5) Inspecting more possible combinations of linear and cyclic alkyl substitutions.

Moreover, ethyl and propyl substitution were also screened. We added detailed discussions in Fig. S2 in the revised supplementary information:

“Fig. S2 | N-doping ability prediction. a-f, Radar charms for predicting n-doping ability of TAMs (a-e) and reported hydride dopants (f). Alkyl substitutions on TAMs include hydrogen/linear alkyls (a), cyclic alkyls (b), fused ring alkyls (c), combinations of rings and methyl groups (d), and 1,5,7-triazabicyclo[4.4.0]decane derivatives (e). Evaluating indexes includes DFT calculated Gibbs free energy change in hydride-transfer half-reaction (ΔG_{H^-}) and hydrogen-atom-transfer half-reaction (ΔG_{T^-}), SOMO level of dopant radical, charge distribution (Mulliken charge or ^1H -NMR chemical shift) of reactive hydrogen, and hydrogen-carbon bond

length (d_{C-H}).

The major structures of TAMs were conceived of by intuitive design and then evaluated. The intuitive design followed simple principle: 1) choosing triaminomethane building block, because it can form a stable guanidine cation after reduction. This comes from our long-time molecular design intuitive and literature research; 2) introducing alkyls to nitrogen atoms to enhance the electron-donating property of TAMs; 3) investigating the effect of methyl and longer alkyl substitutions and evaluate the redox properties and stabilities of the compound by calculation; 4) reducing steric hindrance by inspecting cyclic/fused cyclic alkyl substitutions; 5) inspecting more possible combinations of linear and cyclic alkyl substitutions

Fig. S2(a) illustrates that compared with hydrogens, methyl substitutions on TAMs could obviously enhance the predicted n-doping ability. This can be attributed to the stronger electron-donating property of alkyl than that of hydrogen. Further increasing the alkyl length to ethyl or propyl would lead to decreased predicted n-doping ability. This suggests that ethyl or propyl does not showing obviously stronger electron-donating property than methyl, and while the larger steric hindrances would affect their planarity thus affect their predicted n-doping abilities.

Intuitively, cyclic alkyls would probably show smaller steric hindrances. The predicted n-doping abilities of TAMs with three to six membered aza cyclic alkyls are shown in Fig. S2(b). TAM_{3T} shows unbalanced predicted n-doping ability, indicating that although aziridine is an electron-rich donor but its tension would be unfavorable to stabilize the cation. TAM_{3Q} shows better balanced predicted n-doping ability which is similar to TAM_{Me}, and further increasing ring size to TAM_{3P}/TAM_{3H} would lead to decreased predicted n-doping ability. This suggests that inter-ring steric hindrance in TAM_{3P} and TAM_{3H} may reduce n-doping ability.

Therefore, fused alkyl ring substituted TAMs are inspected to reduce such inter-ring steric hindrance. Fig. S2(c) demonstrates that TAM₅₆₆, TAM and TAM₆₆₇ could have the predicted strongest n-doping ability among all TAMs, and either reducing or increasing ring size may lead to decreased n-doping ability. This indicates that fused aza six-membered rings would be favored to enhance the n-doping ability due to their specific chair conformations and secondary orbital interactions (Fig. S3).

Furthermore, some combinations of rings and methyl substituted TAMs were also investigated. Fig. S2(d) shows that among these combinations, only TAM_{Me66} could have strong n-doping ability comparable to TAM₅₆₆, TAM and TAM₆₆₇. This implies that only the fused aza six-membered rings (1,5,7-triazabicyclo[4.4.0]decane, Fig. S2(e)) backbone might be essential for high predicted n-doping ability in TAM derivatives.”

Q2:

“The facile synthesis and high stability of the TAM-H compound itself are attractive features.

The comparison of hydride transfer activation energies and driving forces between TAM-H and N-DMBI is interesting. However, the high activation energy for TAM-H hydride transfer may not be desirable because a somewhat harsh heating step is required to complete the doping process. Also, the doping is not necessarily via hydride transfer; it may be from homolytic C-H bond cleavage, so the hydride transfer thermodynamics may not be highly relevant. The thermodynamics of doping via hydride or hydrogen atom would be equivalently irreversible for any of the triaminomethanes if H₂ gas were generated or if new, stronger C-H bonds were formed elsewhere in the polymer.”

Our Response:

The efficient TAM n-doping is achieved by annealing TAM/semiconductor blended films at 120~140 °C. In fact, this temperature is also the optimal condition of other n-dopants and polymers systems, e.g.:

N-DMBI/PNDTI-BBT-DP, annealing at 120 °C, *Macromolecules* **50**, 857-864 (2017).

N-DMBI/FBDPPV, annealing at 120 °C, *J. Am. Chem. Soc.* **137**, 6979-6982 (2015).
 N-DMBI/PDPF, annealing at 140 °C, *Adv. Mater.* **30**, 1802850 (2018).
 N-DMBI/N2200, annealing at 150 °C, *Adv. Mater.* **26**, 2825-2830 (2014).
 N-DMBI/P(FBDOPV-2T-C12), annealing at 120 °C, *Adv. Funct. Mater.* 2000449 (2020).
 TBAF/CIBDPPV, annealing at 130 °C, *Adv. Mater.* **29**, 1606928 (2017).

Furthermore, TAM doped FBDPPV, N2200, or PC₆₁BM (annealing at 140 °C) present comparable or even higher electrical conductivity compared with N-DMBI doped ones. Therefore, the higher activation energy of TAM than N-DMBI would not restrict the effective n-doping of polymers.

As the reviewer mentioned, while doping, TAM can be also achieved by homolytic C-H bond cleavage and following electron transfer, the thermodynamics of doping via hydride or hydrogen atom would be equivalently irreversible. We proved that TAM-doped FBDPPV is stable during long time annealing (Fig. S29 in the revised supplementary information). We added a brief discussion about this point in our revised manuscript (Thermoelectric performance with thick polymer films Part):

“Moreover, TAM-doped BDPPV thick films are also stable during long-time thermal annealing in nitrogen (but unstable during thermal annealing in air, see Fig. S29 in supplementary information), indicating that TAM-doping is irreversible.”

Fig. S29 | Stability of electrical conductivity. **a**, Comparison of electrical conductivity stability between TAM-doped FBDPPV thin film (10 nm) and thick film (10 μm) without encapsulation (at their electrical conductivity maxima, under ambient conditions: 25 °C, $R_H = 50\sim 60\%$). **b**, Time-dependent electrical conductivity of TAM-doped FBDPPV thick film under long-term continuous annealing.

Q3:

Thermodynamic parameters are favorable, especially for n-type thin films. However, more insight is needed about why the thick films of this work behave better than in previous works; is the morphology denser, or are the chemical structures inherently more difficult to dedope with oxygen? Is the mobility significantly higher in the doped form compared to other n-doped polymers, or is there some favorable density of states distribution that allows relatively high Seebeck coefficient along with the high conductivity? Figure S21 suggests slightly enhanced mobility from doping, but it is not clear whether this effect is more pronounced than with the N-DMBI dopant.

Our Response:

GIWAXS analysis shows that TAM doped FBDPPV present smaller π - π stacking distances than N-DMBI doped FBDPPV and less disordered π - π stacking. (Fig. 5a-b and Fig. S42d-f in

supplementary information). Thus, TAM doped FBDPPV is denser in molecular π - π stacking. This may lead to enhanced electrical/thermoelectric properties:

1) As the reviewer mentioned, the mobility of TAM doped FBDPPV is obviously higher, this effect is more pronounced than with the N-DMBI. We have proved this in Fig. S24 in the revised supplementary information.

2) TAM doped FBDPPV demonstrates weaker temperature dependence of both electric conductivity and Seebeck coefficient than N-DMBI doped FBDPPV (Fig. S33-S34 in the revised supplementary information).

Fig. S24 | Doping ability. a-b, Transfer characteristics of intrinsic and TAM doped FBDPPV (a) or N-DMBI doped FBDPPV (b) based field-effect transistors (FETs). FET measurements suggest that both TAM and N-DMBI doping can enhance electron mobility of FBDPPV, and this effect is more pronounced in TAM than N-DMBI.

Fig. S33 | Temperature dependent electrical conductivity. a, Temperature dependent electrical conductivity of TAM/N-DMBI doped FBDPPV (at their electrical conductivity maxima). **b,** Evaluation of the nearest-neighbor hopping (NNH) conduction exponent. **c,** Activation energy (W) of TAM/N-DMBI doped FBDPPV in NNH conduction.

Fig. S34 |Temperature dependent Seebeck coefficient. a-b, Temperature dependent Seebeck coefficient of TAM/N-DMBI doped FBDPPV (at their electrical conductivity maxima).

Therefore, the denser molecular π - π stacking in TAM doped FBDPPV may be the key issue to improve the thermoelectric performance. We added this discussion in our revised manuscript (Solid-state microstructures Part):

“The microstructure uniformity could explain the high electrical conductivities of TAM doped FBDPPV in various thicknesses. In contrast, the lower electrical conductivity in the thick films of N-DMBI doped FBDPPV might be probably caused by its inhomogeneous microstructure. More importantly, TAM doping will not reduce but slightly enhance electron mobility of FBDPPV, and this effect is more pronounced than N-DMBI doping (Fig. S24 in supplementary information). TAM doped FBDPPV demonstrates weaker temperature dependence of both electrical conductivity and Seebeck coefficient (Fig. S33-S34 in supplementary information), indicating that TAM doped FBDPPV has lower molecular π - π stacking disorder than N-DMBI doped FBDPPV (Fig. S42d-e in supplementary information).”

Q4:

“The absolute power factor seems about twice as high as other doped polymer systems, based on the authoritative review by one of the coauthors (Zhu) and as stated in the present manuscript. The stability of the electrical conductivity looks promising, but other n-type conducting polymers, such as nickel thiol coordination polymers (Wolfe et al., Adv. Electron. Mater. 2019, 5, 1900066), have been stabilized on similar or greater time scales.”

Our Response:

The nickel-thiol based coordination polymers are an important research direction of n-type thermoelectric polymers. Their air-stability is promising and offering important inspirations. However, solution processability is still the core advantage of polymer thermoelectric materials. We added a brief discussion about this point in our revised manuscript and cited this article as reference 49 (Thermoelectric performance with thick polymer films Part):

“Air-stability in one of the key issues in high-performance n-type thermoelectric materials⁴⁹⁻⁵⁰”

Q5:

“Making a thermoelectric generator is a good demonstration. There is no comparison with other reported thermoelectric generators by which to judge the power output magnitude.”

Our Response:

We added comparison of thermoelectric generator power output as Fig. S53i in the revised supplementary information:

“Fig. S53 | Power output and flexibility of thermoelectric generator. i, Comparison of maximum power output (per number of thermos modules) of some polymer (including coordination polymer) thermoelectric generators with similar device geometries^{S20-S23}.”

We also added a brief discussion about this point in our revised manuscript (All-polymer thermoelectric generator Part):

“Compared to some polymer thermoelectric generators with only PEDOT-PSS p-leg and metal connection lines, this complementary all-polymer generator demonstrates obviously enhanced power output (Fig. S53i in supplementary information).”

Q6:

“Some minor points:

The ZT of 0.42 for PEDOT is overstated because of the geometries of the electronic and thermal conductivity measurements not being the same, and the possible enhancement from ionic conductivity. See Liu et al., *Macromolecules* 2015, 48, 3, 585-591.”

Our Response:

We removed the description of “The ZT of 0.42 for PEDOT”. We modify this as follows in the revised manuscript:

“Doped PEDOT materials exhibited p-type electrical conductivities over 1500 S cm^{-1} , with thermoelectric power factors of up to $469 \mu\text{W m}^{-1} \text{ K}^{-2}$, which demonstrate their application potential compared to commercial inorganic counterparts⁹⁻¹².”

Q7:

“Caption to Figure 2, bottom of page 4: while the TAM⁺ cation may have less interaction with conjugated polymer chains than N-DMBI does, it is not likely to have a “strong” interaction with nonpolar/nonfunctional alkyl chains.”

Our Response:

We removed the description of “strong interaction with alkyl chains”. We modify this as follows in our revised manuscript:

“TAM⁺ cation has distinctly weaker interactions with polymer backbones, while NDMBI⁺ cation presents stronger interactions with polymer backbones.”

For Reviewer #2:

The manuscript by Yang, et al. reports a new n-type dopant based on a triaminomethane derivative. The manuscript is thorough and systematically goes through the design of the dopant family, including analysis of a number of calculated parameters relevant to their doping behavior, and then reports the synthesis and doping behavior of one of the predicted high-performing derivatives. The new dopant, TAM, is shown to have high stability and is used to dope FBDPPV and achieve high n-type thermoelectric power factors, including in much thicker films than are accessible with the dopant N-DMBI. The manuscript provides a nice comparison between TAM and N-DMBI throughout. The development of this new n-type dopant, which may potentially lead to a larger family of n-type dopants, is significant and the work is detailed and thorough. It would be helpful if the authors can address the following concerns:

Q1:

“In Figure 1c, it is hard to distinguish some of the colors, for example TAM555 and TAM. I recommend using the same color scheme, but using differing symbols for each TAM derivative. The symbols should be placed to indicate the intersection with each axis. This would allow the derivatives to be more easily identified.”

Our Response:

We split Fig. 1c to new Fig. 1c and Fig. S2a-e in supplementary information to make them clear. We also placed symbols for each TAMs derivative to indicate the intersection with each axis in the new Fig. 1c and Fig. S2a-e in supplementary information as the reviewer suggested.

Fig. 1 | Dopant design and n-doping ability prediction. **c-d**, Radar charms for predicting n-doping ability of TAMs (**c**) reported hydride dopants (**d**). Evaluating indexes includes DFT calculated Gibbs free energy change in hydride-transfer half-reaction (ΔG_{H^-}) and hydrogen-atom-transfer half-reaction (ΔG_{H^\bullet}), SOMO level of dopant radical, charge distribution (Mulliken charge or $^1\text{H-NMR}$ chemical shift) of reactive hydrogen, and hydrogen-carbon bond length ($d_{\text{C-H}}$).

“Fig. S2 | N-doping ability prediction. a-f, Radar charms for predicting n-doping ability of TAMs (a-e) and reported hydride dopants (f). Alkyl substitutions on TAMs include hydrogen/linear alkyls (a), cyclic alkyls (b), fused ring alkyls (c), combinations of rings and methyl groups (d), and 1,5,7-triazabicyclo[4.4.0]decane derivatives (e). Evaluating indexes includes DFT calculated Gibbs free energy change in hydride-transfer half-reaction (ΔG_{H^-}) and hydrogenatom-transfer half-reaction (ΔG_{H^\bullet}), SOMO level of dopant radical, charge distribution (Mulliken charge or $^1\text{H-NMR}$ chemical shift) of reactive hydrogen, and hydrogen-carbon bond

length (d_{C-H}).

Q2:

In the radar charts it is not initially clear whether the values (e.g., 5.5 to 1.5) are listed in order from the outside to the center or vice versa. Clearly indicating whether the first value listed is the center or edge of the axis would be helpful.

Our Response:

We added labels of outside/center in each radar charts (Fig. 1c-d, Fig. S2a-f in supplementary information, see above) as the reviewer suggested.

Q3:

“The end of the Figure 1 caption says “TAM and TAM556 show the highest n-doping ability among TAMs.” I think that TAM₅₅₆ should be TAM₅₆₆ and this should say the highest predicted n-doping ability. The statement as written led me to expect that these derivatives would all be experimentally synthesized and characterized, which is not the case. Only TAM is reported.”

Our Response:

We revised this description to “TAM and other TAMs with 1,5,7-triazabicyclo[4.4.0]decane backbone (TAM₅₆₆, TAM₆₆₇ and TAM_{Me66}) show the highest predicted n-doping ability among TAMs.” as the reviewer suggested.

We also clarified that only TAM is experimentally synthesized and characterized. We added this in our revised manuscript (Synthesis, stability, and doping kinetics Part)

“Although TAM and other TAMs with 1,5,7-triazabicyclo[4.4.0]decane backbone (TAM₅₆₆, TAM₆₆₇ and TAM_{Me66}) are predicted to be strong hydride n-dopants, only TAM is synthesized in this work.”

Q4:

“Why are the lamellar stacking distances lengthened from the observed 32.9 Å to 50 Å for the MD simulations? Lengthening this spacing introduces significantly more free volume which is likely going to lead to significant differences in the MD simulations as compared to if the experimentally observed lamellar stacking distances were used. This may be one of the reasons that the backbones are less planar in the TAM containing system than the N-DMBI system. This increased disorder in the conjugated backbone observed in the simulation for TAM is not apparent in the experimental GIWAXS measurements.”

Our Response:

As we mentioned in manuscript, lengthening lamellar packing distances to 50 Å makes the system become non-interdigitated (following the same method from Nature **515**, 384-388 (2014)). Using interdigitated initial structure (e.g. 32.9 Å) without free volumes would need to immobile dopant cations during MD simulations. The MD simulated molecular interactions should not depend on initial packing distances. To prove this, we added an additional MD simulation with smaller initial lamellar distance of 42 Å. This MD simulation shows consistent results on dopant cation-polymer intermolecular interactions (Fig. S7-S8 in supplementary information): statistical analysis shows that TAM⁺ cations pervasively move away from the backbone toward middle of alkyl sidechains, while N-DMBI⁺ cations move toward the backbone or tails of alkyl sidechains.

We added following discussions in the revised manuscript:

“Additional MD simulation with a different initial supercell size (lamellar distance = 42 Å, other conditions keep the same) of TAM/N-DMBI doped FBDPPV shows consistent results (Fig. S8 in supplementary information), suggesting the simulation results do not depend on the supercell sizes.”

Fig. S7 | Cation-sidechain interaction prediction. a-f, Histogram of distances between dopant cations and polymer conjugated backbones for TAM doped FBDPPV (a, c, e) and N-DMBI doped FBDPPV (b, d, f) with initial lamellar distance of 50 Å. Notice distance = 0 is polymer backbone and distance = 25 Å is alkyl chain tails.

Fig. S8 | Cation-sidechain interaction prediction. a-f, Histogram of distances between dopant cations and polymer conjugated backbones for TAM doped FBDPPV (a, c, e) and N-DMBI doped FBDPPV (b, d, f) with initial lamellar distance of 40 Å. Notice distance = 0 is polymer backbone and distance = 21 Å is alkyl chain tails.

Q5:

“At the end of page 5 the text states: “the hydride-transfer reaction of TAM is slower than that of N-DMBI (Fig. 5d and Fig. 5g).” This text is not referring to the correct figures. Additional incorrect figure reference appears later in that same paragraph at the top of page 6.”

Our Response:

We revised these incorrect figure references in our revised manuscript:

“the hydride-transfer reaction of TAM is slower than that of N-DMBI (Fig. 5c and Fig. S17 in supplementary information).”

“Hydride-transfer transition states calculations on other substituted tritylium ions and other typical electrophiles show that TAM has consistently higher ΔG_{298K}^\ddagger and lower ΔG_{298K} in hydride-transfer reactions (Fig. S18-S19 in supplementary information).”

Q6:

“The statement “UPS measurement shows that the secondary electron cutoffs of both FBDPPV and N2200 shift obviously to lower kinetic energies” is odd, as the secondary electron cutoff is looking at the electrons with near 0 kinetic energy. These kinetic energies are not changing. The authors likely mean that the secondary electron cutoff shifts to higher binding energies, indicating a decreased work function.”

Our Response:

We revised this description to in our revised manuscript as the reviewer suggested:

“UPS measurement shows that the secondary electron cutoffs of both FBDPPV and N2200 shift obviously to higher binding energies, indicating decreased work functions after the effective TAM n-doping (Fig. S23a-d in supplementary information).”

Q7:

“From the ultraviolet photoelectron spectra in Figure S20 the HOMO onset of the polymers cannot be seen due to the large y-scale used. A zoomed in view of the HOMO onset region should be included.”

Our Response:

We revised Fig.S20 in the revised supplementary information as the reviewer suggested:

Fig. S23 | Doping ability. a-d, Ultraviolet photoelectron spectra of intrinsic and TAM doped FBDPPV (a, c) and N2200 (b, d) thin films at low kinetic energy region (a-b) and low binding energy region (c-d).”

Q8:

“On page 7 the authors say “as merit ZT”. This should say “as the figure of merit ZT”. The authors should define both ZT and the power factor.”

Our Response:

We revised this description in our revised manuscript as the reviewer suggested:

“Geometry and thickness of thermolegs also play an important role in power conversion efficiency and power output maximum as well as the figure of merit ZT and power factor of thermoelectric materials⁴⁷.”

We define ZT and the power factor in our revised manuscript (Thermoelectric performance with thick polymer films Part):

“Thermoelectric properties of polymers can also be evaluated in terms of the figure of merit ZT or power factor (PF):

$$ZT = \frac{S^2 \sigma}{\kappa} T \quad (1)$$

$$PF = S^2 \sigma \quad (2)$$

Where σ is electrical conductivity (S cm⁻¹), S is Seebeck coefficient (μV K⁻¹), and κ is thermal conductivity (W m⁻¹K⁻¹) of a polymer.”

Q9:

“Plot 4c should be plotted with electrical conductivity on a linear scale to allow for changes to be more easily observed.”

Our Response:

We revised Fig. 4c with electrical conductivity on a linear scale as the reviewer suggested.

Fig. 4 | Thermoelectric performance of TAM doped organic semiconductors. c, Air stability of TAM-doped FBDPPV thick films without encapsulation.

Q10:

“The discussion of the actual thermoelectric module is not too informative. Neither the figure nor discussion contain any dimensions and no measurement or even an estimate of the temperature gradient across the device is presented.”

Our Response:

We add detailed discussions about the actual thermoelectric module in both revised manuscript and supplementary information as the reviewer suggested. In manuscript:

“The generator shows a maximum power output of 0.18 nW at $T_{hot} = 30\text{ }^{\circ}\text{C}$ (with temperature gradient $\Delta T = 2.3\text{ K}$), and the maximum power output reaches 25 nW and 77 nW after T_{hot} increasing to 100 $^{\circ}\text{C}$ and 150 $^{\circ}\text{C}$ ($\Delta T = 27.8\text{ K}$ and 46.5 K, respectively), even though the generator only has an air heat sink and its thermoleg geometry was not fully optimized (Fig. 6f, Fig. S53a-f in supplementary information). The thermoelectric generator also shows stable power output during bending its thermolegs (bending radius = 1 cm, Fig. S53g-h in supplementary information). Compared to some polymer thermoelectric generators with only PEDOT-PSS p-leg and metal connection lines, this complementary all-polymer generator demonstrates obviously enhanced power output (S53i in supplementary information).”

In supplementary information:

“Fig. 53 | Power output and flexibility of thermoelectric generator. a, Measuring method of temperature gradient ($\Delta T = T_1 - T_2$) on thermolegs, T_2 and T_1 are temperatures of the top and end of thermolegs, respectively. T_1 and T_2 are measured using thermocouples which are stucked tightly on thermolegs with kapton tape. **b-c**, Temperature gradient on thermolegs with different hot plate temperature. **d-e**, Output voltage and power of the generator at different temperature gradient. **f**, Maximum power output of the generator at different temperature gradient. **g**, Photograph showing bending radius of the legs of thermoelectric generator. **h**, Power output of the thermoelectric generator during 20 bending cycles (bending radius = 1 cm). **i**, Comparison of maximum power output (per number of thermos modules) of some polymer (including coordination polymer) thermoelectric generators with similar device geometries^{S20-S23}. The all-polymer thermoelectric generator presents good flexibility with high power output.”

Q11:

“It would be helpful if the authors clearly stated early on in the paper that only TAM was synthesized. After reading through the theory section I was expecting multiple derivatives to be made and reported. Reporting only TAM is enough, but identifying this early on would be helpful to better direct expectations.”

Our Response:

We clarified that only TAM is experimentally synthesized and characterized as the reviewer suggested. We added this in the revised manuscript (Synthesis, stability, and doping kinetics Part)

“Although TAM and other TAMs with 1,5,7-triazabicyclo[4.4.0]decane backbone (TAM₅₆₆, TAM₆₆₇ and TAM_{Me66}) are predicted to be strong hydride n-dopants, only TAM is synthesized in this work.”

For Reviewer #3:

The authors report on the synthesis of a new n-type dopant named TAM (based on triaminomethane) with high stability and strong thermally activated doping mechanism. TAM was then integrated into operational thermoelectric materials as a dopant, enabling creation of a thick (up to 10 mm) films with high output voltage and current.

The idea is to obtain the n-type dopant without all the conventional issues like stability and miscibility, which typically affect organic thermoelectric systems, and further use it for n-type doping.

The topic is highly interesting and deals with the complex and advanced approach of dopant formation, involving theoretical DFT-based calculations, in parallel to experiments. The target is to investigate the influence of the dopant in lamellar and p-p stacking of the targeted for doping polymers. The field of direct transformation of heat into electricity through the thermoelectric effect is fast evolving and attracts ever growing interest.

One of the main weaknesses in the field is the lack of n-type materials and thus the research presented in the manuscript is very timely and interesting. It can be of interest for the broad audience of Nature Communication, not only involved in the field of thermoelectric, but also interested in the field of organic electronics.

While the idea reported by the authors is original and appealing, there are some statements that should be addressed. A detailed analysis on the weaknesses of the manuscript is reported below.

On the basis of the observations below, I would suggest the authors to modify the manuscript according, before reconsidering it for publication.

Overall, I would suggest major revisions before accepting the manuscript for publication on Nature Communications.

Q1:

“The authors state that TAM was discovered. They should clarify: was TAM firstly synthesized or firstly applied as a dopant?”

Our Response:

We revised this description of “discovered” into “developed” in our revised manuscript.

We also clarified that TAM was firstly used as a dopant in our revised manuscript (Conclusions Part):

“TAM is also firstly used as an n-dopant, and TAM presents several unique features”

Q2:

“Please consider the following publication with much higher values of power factor "ACS Appl. Mater. Interfaces 2019, 11, 3400–3406””

Our Response:

The MOF/ions based materials are an important research direction of n-type thermoelectric materials. Their air-stability is promising and offering important inspirations. However, solution processability is still the core advantage of polymer thermoelectric materials. We added a brief discussion about this point in our revised manuscript and cited this article as reference 50 (Thermoelectric performance with thick polymer films Part):

“Air-stability in one of the key issues in high-performance n-type thermoelectric materials⁴⁹⁻⁵⁰”

Q3:

“What parameters were used to calculate the DFT. Spin-polarized calculation? Spin-orbital coupling considered? +U parameter? Hybrid pseudopotentials? The authors should clarify this point.”

Our Response:

Since all DFT calculations in this work only involved molecules with very light elements (C, H, O, N, F) and simple spin systems not exceeding single radical, general hybrid functional (B3LYP, ω B97XD) were used. Neither of spin-polarized calculation, spin-orbital coupling, +U parameter, or hybrid pseudopotentials was involved. We added detailed description of DFT calculation methods in our revised manuscript (Methods Part) as the reviewer suggested:

“Density functional theory calculations. Geometry optimizations and n-doping ability evaluating indexes calculation on TAMs and other hydrides were performed under B3LYP/6-311+G(d,p) level. For dopant radicals, uB3LYP/6-311+G(d,p) was employed. For isotropic polarizability of dopant cations, B3LYP/6-311+G(d,p) were used. Hydride-transfer transition state calculations were performed under ω B97XD/6-311+G(d,p)//B3LYP/6-31G(d) level. For absorption spectra calculations, geometry optimization on FBDPPV trimer, trimer anion and trimer dianion were performed under B3LYP/6-31G(d) level. Time-dependent density functional theory (TD-DFT) calculations on vertical excitation energy were performed under B3LYP/6-31G(d) level. For trimer anion, uB3LYP/6-311+G(d,p) were employed.”

Q4:

“The author should consider to split or enlarge Fig 3. In its present form, it is too difficult to read.”

Our Response:

We split Fig. 3 to new Fig. 3 and Fig. S15, S17a, S18c in supplementary information to make them clear as the reviewer suggested.

Fig. 3 | Synthesis, chemical property, and n-doping kinetics. **a**, Synthesis, chemical stability, and hydride-transfer ability of TAM. **b**, Time-dependent ¹H-NMR spectra of TAM in boiling H₂O-d₂. **c**, Time-dependent absorbance of hydride-transfer reaction between (MeO)₃Tr⁺ and TAM (or N-DMBI). The initial concentrations are 2.38×10^{-5} M for (MeO)₃Tr⁺, 1.42×10^{-4} M for both TAM and N-DMBI in anhydrous dichloromethane. **d**, DFT calculated activation ΔG_{298K}^\ddagger , reaction ΔG_{298K} , and energy surface for hydride-transfer reaction between (MeO)₃Tr⁺ and TAM (or N-DMBI). DFT calculation performed under ω B97XD/6-311+G(d,p)//B3LYP/6-31G(d) level. **e-f**, Absorption spectra of intrinsic and TAM (**e**) or N-DMBI (**f**) doped FBDPPV dilute ODCB solution. The polymer concentration keeps 20 mg/L (in ODCB) for each case, with different mass

fraction of dopants after mixing for 5 min. TAM has ultra-high stability with strong n-doping ability at high temperature.

Fig. S15 | Chemical stability. Time-dependent ^1H NMR spectra of TAM in saturated DCI-*d* ($\text{D}_2\text{O}-d_2$ solution). Notice that ^1H NMR spectra of TAM are totally different from that of $\text{TAM}^+\text{BF}_4^-$ in the same conditions (See Part 9. ^1H and ^{13}C NMR spectra), indicating that TAM cannot transfer hydride to proton to generate hydrogen in boiling hydrochloric acid. Therefore, TAM is stable even in boiling hydrochloric acid.

Fig. S17 | Hydride-transfer reaction rates. **a**, Absorption spectra of intrinsic and TAM (or N-DMBI) mixed $(\text{MeO})_3\text{Tr}^+$ after mixing for 5 min.

Fig. S18 | Kinetic nucleophilicity. **c**, DFT calculated activation Gibbs free energy (ΔG^\ddagger) and reaction Gibbs free energy (ΔG) of the hydride-transfer reactions.

Q5:

“Absorption spectra shows that N-DMBI can n-dope FBDPPV at room temperature,” How exactly authors estimate it from absorption spectra? Which peak refers to what energy state and how electrons are connected? This aspect is not clear.

Our Response:

We added time-dependent density functional theory (TD-DFT) calculations to clarify the absorption features after n-doping. TD-DFT calculations reveal that in TAM doped FBDPPV (140 °C) and N-DMBI doped FBDPPV (room temperature), the absorption bands (900 to 1400 nm, Band III; 1400 to 2000 nm, Band IV) at long wavelength are distinct from the absorption of intrinsic FBDPPV (400 to 650 nm, Band I; 650 to 900 nm, Band II), which can be attributed to polaronic absorption of negative charged FBDPPV polymer. Furthermore, Band III might be attributed to the absorption of polaron anions, and Band IV might be attributed to the absorption of bipolaron dianions. We added this discussion in Fig. S22 in the revised supplementary information:

Fig. S22 | Experimental and calculated absorption spectra. **a**, Experimental absorption spectra of intrinsic and doped FBDPPV in dilute ODCB solution at room temperature. The FBDPPV concentration keeps 20 mg/L for each case, after mixing with TAM (at 140 °C for 5 min) or N-DMBI (at room temperature for 5 min). **b**, Time-dependent density functional theory (TD-DFT) calculated absorption spectra and oscillator strength of FBDPPV trimer, (FBDPPV trimer)⁻ anion, and (FBDPPV trimer)²⁻ dianion. The geometry configuration of FBDPPV trimer, (FBDPPV trimer)⁻ anion, and (FBDPPV trimer)²⁻ dianion were firstly optimized, then the first 50 vertical excitation energy (ΔE_{ve}) and their corresponding oscillator strength of each compound were calculated (Table S4-S6). Finally, the absorption spectra were generated using Gaussian peak fitting with half-width at half height of 0.10 eV for each peak. Geometry optimizations and TD-DFT calculations were under B3LYP/6-31G(d) level, long alkyl chains were replaced with methyl to simplify the calculation.

TD-DFT calculations^{S16} reveal that in TAM doped FBDPPV (140 °C) and N-DMBI doped FBDPPV (room temperature), the absorption bands (900 to 1400 nm, Band III; 1400 to 2000 nm, Band IV) at long wavelength are distinct from the absorption of intrinsic FBDPPV (400 to 650 nm, Band I; 650 to 900 nm, Band II), which can be attributed to polaronic absorption of negatively charged FBDPPV polymer. Furthermore, Band III might be attributed to the absorption of polaron anions, and Band IV might be attributed to the absorption of bipolaron dianions.”

Q6:

““The doped polymer thin films (10 nm) were fabricated through spin-casting from ODCB” Can authors show the SEM or TEM images exhibiting the thickness of the doped polymer?”

Our Response:

We added the SEM image showing the thickness of the TAM doped FBDPPV in Fig. S51 in the revised supplementary information as the reviewer suggested:

Fig. S51 | SEM analysis. Scanning electron microscope (SEM) image recorded at 10.00 kV acceleration voltage showing the cross section of TAM doped FBDPPV (5% TAM, at the electrical conductivity maximum, 10 μm) drop-casted on a Si/SiO₂ (1 μm SiO₂) substrate. The SEM analysis shows that TAM doped FBDPPV is a compact film.”

Q7:

““TAM-doped FBDPPV thick films are stable in air.” is that statement valid at elevated temperatures? With increase of T , the TAM doped films are supposed to exhibit higher S , since TAM is thermally activated, thus changing the chemistry of TAM layer on top of the FBDPPV. Please clarify.”

Our Response:

TAM doped FBDPPV thick films are not stable at high temperature (e.g. 100 °C), due to significantly enhanced oxygen and water diffusion. We clarified this in both revised manuscript and supplementary information as the reviewer suggested.

In manuscript:

“Moreover, TAM-doped BDPPV thick films are also stable during long time thermal annealing in nitrogen (but unstable during thermal annealing in air, see Fig. S29 in supplementary information)”

In supplementary information:

Fig. S29 | Stability of electrical conductivity. **a**, Comparison of electrical conductivity stability between TAM-doped FBDPPV thin film (10 nm) and thick film (10 μm) without encapsulation (at their electrical conductivity maxima, under ambient conditions: 25 °C, $R_H = 50\sim 60\%$). **b**, Time-dependent electrical conductivity of TAM-doped FBDPPV thick film under long-term continuous annealing.

Time-dependent electrical conductivity measurement shows that TAM-doped FBDPPV thin films are unstable in air, while the thick film is stable in air. This indicates that TAM-doped FBDPPV thick film can form self-encapsulating to prevent bulk oxidative de-doping under room temperature. Long-term continuous annealing measurement shows that TAM doped FBDPPV thick film is also stable at high temperature under nitrogen. This would imply that TAM doping is stable and irreversible. Notice that TAM-doped FBDPPV thick film is unstable during heating in air, suggesting that the self-encapsulating is invalid at high temperature. CYTOP was employed to encapsulate TAM-doped FBDPPV thick film. However, the CYTOP encapsulate layer can only slow down but cannot prevent the oxidative de-doping. Thus, more effective encapsulation methods need to be developed in the future studies.”

We proved that with increasing of temperature (T), the TAM doped FBDPPV are supposed to exhibit higher Seebeck coefficient (S) and electrical conductivity (σ) (Fig. S33-S34). We clarified this in both revised manuscript and supplementary information as the reviewer suggested.

In manuscript:

“The microstructure uniformity could explain the high electrical conductivities of TAM doped FBDPPV in various thicknesses. In contrast, the lower electrical conductivity in the thick films of N-DMBI-doped FBDPPV might be probably caused by its inhomogeneous microstructure. More importantly, TAM doping will not reduce but slightly enhance the electron

mobility of FBDPPV, and this effect is more pronounced than N-DMBI doping (Fig. S24 in supplementary information). TAM doped FBDPPV demonstrates slightly weaker temperature dependence of both electric conductivity and Seebeck coefficient (Fig. S33-S34 in supplementary information). This indicates that TAM doped FBDPPV may have lower molecular π - π stacking disorder than N-DMBI doped FBDPPV (Fig. S42d-e in supplementary information).”

“**Fig. S33 | Temperature dependent electrical conductivity.** **a**, Temperature dependent electrical conductivity of TAM/N-DMBI doped FBDPPV (at their electrical conductivity maxima). **b**, Evaluation of the nearest-neighbor hopping (NNH) conduction exponent. **c**, Activation energy (W) of TAM/N-DMBI doped FBDPPV in NNH conduction.

Both TAM and N-DMBI doped FBDPPV shows slightly higher electrical conductivity at higher temperature (Fig. S33a). This thermally activated conduction could be described by the nearest-neighbor hopping (NNH) mechanism (Fig. S33b)^{S12-S14}. In this NNH conduction, both TAM and N-DMBI doped FBDPPV show weak temperature dependent electrical conductivity with small activation energy of 23.6 meV and 33.4 meV, respectively (Fig. S33c). Notice that TAM doped FBDPPV shows slightly weaker temperature dependence of electrical conductivity than N-DMBI doped FBDPPV. This indicates that TAM doped FBDPPV would have lower molecular disorder in solid state^{S15-S17}, which is consistent with the GIWAXS analysis.

Fig. S34 | Temperature dependent Seebeck coefficient. **a-b**, Temperature dependent Seebeck coefficient of TAM/N-DMBI doped FBDPPV (at their electrical conductivity maxima).

Both TAM and N-DMBI doped FBDPPV shows slightly higher Seebeck coefficient at higher temperature (Fig. S34a). Notice that TAM doped FBDPPV shows slightly weaker temperature dependence of Seebeck coefficient ($\partial S/\partial(1/T) = 14.9$ meV) than N-DMBI doped FBDPPV ($\partial S/\partial(1/T) = 27.1$ meV). This indicates that TAM doped FBDPPV could have lower molecular disorder in solid state^{S15-S17}, which is consistent with GIWAXS analysis.”

Q8:

“Output voltage should be converted from mW (which is not at all a voltage) to mV for the better comparison with other research results.”

Our Response:

We revised this point, and changed mW to mV in Fig. 6f.

Fig. 6 | Flexible all-polymer solution-processed thermoelectric generator. f, Output voltage and power of the generator at different heat source temperature (air temperature is 25 °C).

We also added comparison with other research results as Fig. S53i in the revised supplementary information:

“Fig. S53 | Power output and flexibility of thermoelectric generator. i, Comparison of maximum power output (per number of thermos modules) of some polymer (including coordination polymer) thermoelectric generators with similar device geometries^{S20-S23}.”

We added this discussion in the revised manuscript:

“Compared to some polymer thermoelectric generators with only PEDOT-PSS p-leg and metal connection lines, this complementary all-polymer generator demonstrate obviously enhanced power output (S53i in supplementary information).”

Q9:

If authors emphasize the flexibility of the generators then the appropriate test should be conducted, i.e., the influence of folding times on TE properties. Additionally, the folding of the "legs" is shown in a way where the actual p- and n-type materials are not changing their shape whereas the substrate is bending.

Our Response:

We revised this description from “flexible thermoelectric generator” to “all-polymer thermoelectric generator” in our revised manuscript. We also added bending stability experiments as Fig. S53g-h in the revised supplementary information:

“Fig. S53 | Power output and flexibility of thermoelectric generator. g, Photograph showing bending radius of the legs of thermoelectric generator. **h,** Power output of the thermoelectric generator during 20 bending cycles (bending radius = 1 cm).”

We added this discussion in our revised manuscript:

“The thermoelectric generator also shows stable power output during bending tests (bending radius = 1 cm, Fig. S53g-h in supplementary information).”

Q10:

“Output parameters (voltage, current, power) are all acquired at T differences from 100 to 150 °C which are too far from differences that can be reached in real life. The authors should provide the same parameters at difference of 30 °C which is more realistic, or explain why this is not possible.”

Our Response:

In waste heat recovery systems (e.g car waste heat), heat source temperatures of 100 to 150 °C are reasonable (*Appl. Therm. Eng.* **101**, 490-495 (2016).). Moreover, this thermoelectric generator not only works under 100 to 150 °C, it can also work at 30 °C or 35 °C (with lower power output). We added this discussion in both revised manuscript and supplementary information as the reviewer suggested.

In manuscript:

“The generator shows a maximum power output of 0.18 nW at $T_{\text{hot}} = 30$ °C (with temperature gradient $\Delta T = 2.3$ K), and the maximum power output reaches 25 nW and 77 nW after T_{hot} increasing to 100 °C and 150 °C ($\Delta T = 27.8$ K and 46.5 K, respectively), even though the generator only has an air heat sink and its thermoleg geometry was not fully optimized (Fig. 6f, Fig. S53a-f in supplementary information).”

In supplementary information:

“**Fig. S53 | Power output and flexibility of thermoelectric generator.** **a**, Measuring method of temperature gradient ($\Delta T = T_1 - T_2$) on thermolegs, T_2 and T_1 are temperatures of the top and end of thermolegs, respectively. T_1 and T_2 are measured using thermocouples which are stucked tightly on thermolegs with kapton tape. **b-c**, Temperature gradient on thermolegs with different hot plate temperature. **d-e**, Output voltage and power of the generator at different temperature gradient.”

Reviewers' Comments:

Reviewer #1:

Remarks to the Author:

The questions were answered thoughtfully, carefully, and comprehensively. The paper is now suitable for acceptance in Nature Communications.

Reviewer #2:

Remarks to the Author:

The authors have sufficiently addressed all of my original concerns and the article is now suitable for publication.

Reviewer #3:

Remarks to the Author:

The authors did a thorough job to address the criticisms. The comments were properly responded, and authors have provided more evidence of their statements.

However, two criticisms were not fully addressed.

Question number 2:

"Please consider the following publication with much higher values of power factor "ACS Appl. Mater. Interfaces 2019, 11, 3400–3406"

I suggested to compare result of PF from the literature with the results in the manuscript. The authors claim that their work is superior to all the others, whereas the above mentioned work shows higher values.

Question 10.:

"Output parameters (voltage, current, power) are all acquired at T differences from 100 °C to 150 °C which are too far from differences that can be reached in real life. Can authors provide the same parameters at difference of 30 C which is more realistic."

Authors indeed provided the output voltage for tests with lower T_{hot} , which makes their answer appropriate. However, during the discussion of the question, they have emphasized that T of 100 °C and 150 °C are plausible to test at, which is contradictory with what they were saying on the response to question number 7.

The text reports: "TAM doped FBDPPV thick films are not stable at high temperature (e.g. 100 °C), due to significantly enhanced oxygen and water diffusion".

Why the authors provide the output parameters at Temperatures above the declared stability?

I would suggest these two minor revisions before accepting the manuscript for publication.

For Reviewer #1:

The questions were answered thoughtfully, carefully, and comprehensively. The paper is now suitable for acceptance in Nature Communications.

For Reviewer #2:

The authors have sufficiently addressed all of my original concerns and the article is now suitable for publication.

For Reviewer #3:

The authors did a thorough job to address the criticisms. The comments were properly responded, and authors have provided more evidence of their statements. However, two criticisms were not fully addressed.

Q1:

“Question number 2:

“Please consider the following publication with much higher values of power factor "ACS Appl. Mater. Interfaces 2019, 11, 3400–3406"

I suggested to compare result of PF from the literature with the results in the manuscript. The authors claim that their work is superior to all the others, whereas the above mentioned work shows higher values.”

Our Response:

In the literature "ACS Appl. Mater. Interfaces 2019, 11, 3400–3406", K.-L. Lu *et al* reported a Zr-MOF/Polyaniline/PSS composite film with n-type electrical conductivity of 0.021 S cm^{-1} , high Seebeck coefficient of $-17780 \mu\text{V K}^{-1}$, and high power factor of $664 \mu\text{W m}^{-1} \text{ K}^{-1}$. However, this Zr-MOF/Polyaniline/PSS composite involves metal, and the large Seebeck coefficient and large power factor are mainly contributed by ionic Soret effect (the authors emphasized this ionic Soret twice in the literature). It is well known that for an ionic thermoelectric system, the ions cannot pass through the interface of the thermoelectric materials and electrode, thus leading to only transient current and no continuous power output (Crispin, X. *et al. Adv. Elec. Mater.* **3**, 1700013 (2017).). Here, we compare thermoelectric materials that based on electron transport and can provide continuous power output. Therefore, the Zr-MOF/Polyaniline/PSS composite, though showing much higher values of power factor, cannot be simply compared with our system.

Q2:

“Question 10:

“Output parameters (voltage, current, power) are all acquired at T differences from $100 \text{ }^\circ\text{C}$ to $150 \text{ }^\circ\text{C}$ which are too far from differences that can be reached in real life. Can authors provide the same parameters at difference of $30 \text{ }^\circ\text{C}$ which is more realistic.”

Authors indeed provided the output voltage for tests with lower T_{hot} , which makes their answer appropriate. However, during the discussion of the question, they have emphasized that T of $100 \text{ }^\circ\text{C}$ and $150 \text{ }^\circ\text{C}$ are plausible to test at, which is contradictory with what they were saying on the response to question number 7.

The text reports: “TAM doped FBDPPV thick films are not stable at high temperature (e.g. $100 \text{ }^\circ\text{C}$), due to significantly enhanced oxygen and water diffusion”.

Why the authors provide the output parameters at Temperatures above the declared stability? I would suggest these two minor revisions before accepting the manuscript for publication.”

Our Response:

Although the FBDPPV-TAM system is unstable when completely exposed to air and heated above 100 °C, it is still stable at high temperature in nitrogen, and it also has relative stability after being encapsulated by CYTOP in air and heated above 100 °C (see Supplementary Figure 29). Therefore, we used CYTOP to encapsulate the all-polymer thermoelectric generator and the devices worked well in air at high temperatures up to 100~150 °C. In addition, these experiments also suggest that our devices are stable under heating, and the instability is largely due to the oxygen/water penetration, which can be overcome by using appropriate encapsulation materials.